# Improving Three-Dimensional Building Segmentation on Three-Dimensional City Models through Simulated Data and Contextual Analysis for Building Extraction

Frédéric Leroux [1,*] , Mickaël Germain [1] , Étienne Clabaut [1] , Yacine Bouroubi [1] and Tony St-Pierre [2]

1   Department of Applied Geomatics, Center for Applications and Research in Remote Sensing (CARTEL), University of Sherbrooke, 2500 Boulevard de l'Université, Sherbrooke, QC J1K 2R1, Canada; mickael.germain@usherbrooke.ca (M.G.); etienne.clabaut@usherbrooke.ca (É.C.); yacine.bouroubi@usherbrooke.ca (Y.B.)
2   XEOS Imaging Inc., 1405 Boulevard du Parc-Technologique, Bureau 110, Quebec City, QC G1P 4P5, Canada
*   Correspondence: lerf2103@usherbrooke.ca

**Abstract:** Digital twins are increasingly gaining popularity as a method for simulating intricate natural and urban environments, with the precise segmentation of 3D objects playing an important role. This study focuses on developing a methodology for extracting buildings from textured 3D meshes, employing the PicassoNet-II semantic segmentation architecture. Additionally, we integrate Markov field-based contextual analysis for post-segmentation assessment and cluster analysis algorithms for building instantiation. Training a model to adapt to diverse datasets necessitates a substantial volume of annotated data, encompassing both real data from Quebec City, Canada, and simulated data from Evermotion and Unreal Engine. The experimental results indicate that incorporating simulated data improves segmentation accuracy, especially for under-represented features, and the DBSCAN algorithm proves effective in extracting isolated buildings. We further show that the model is highly sensible for the method of creating 3D meshes.

**Keywords:** semantic segmentation; 3D building segmentation; cluster analysis; 3D city models; 3D mesh; data simulation; contextual analysis; Markov random fields; Unreal Engine; Evermotion

## 1. Introduction

The concept of three-dimensional (3D) cities and digital twins is gaining popularity in the representation of natural and human environments [1–3]. A digital twin is a virtual replica of a real physical entity or system, including its surroundings and associated processes. Consistently refreshed through the exchange of information with its real-world counterpart, it can be represented in a virtual environment, facilitating various simulations for comprehensive analysis and evaluation. This concept is attracting interest in many fields to solve problems of multidisciplinary nature, such as efficient input management in agriculture [4,5], building design and management [6,7] and even problems in the healthcare sector [8,9]. A 3D city is a popular type of digital twin in the urban sector. This tool has versatile applications, serving as a decision support tool in urban planning, route planning, natural disaster management, virtual tourism, and the estimation of emergency response times. The incorporation of 3D cities enables the visualization of objects in a virtual environment, allowing users to access detailed information such as the type of use, surface area, year of construction, and more [1]. For example, [10] have produced a 3D model of a Greek city whose surfaces have been divided by pavement type with the aim of enabling the city to perform fire propagation simulations in the event of a fire. This type of process can be carried out automatically using semantic segmentation and instance segmentation. The first process entails categorizing a 3D model into various classes of objects, whereas the second involves identifying and isolating each segmented object

into separate elements. Artificial intelligence (AI) provides many possibilities with deep learning using neural networks. According to the literature, the use of convolutional neural networks (CNNs) delivers the best segmentation results [7,11–16].

Three-dimensional models have the advantage of possessing geometric information and spectral information from the red, green, and blue (RGB) bands. The integration of spectral information can improve segmentation by, for example, using surface color to differentiate objects [11–17]. This information is very important for separating objects that have obvious color discontinuities but great geometric continuity, such as houses arranged in rows that have different siding color. To accomplish this, a large amount of annotated data is needed to train a neural network. It is in this context that data simulation is used extensively. Indeed, work has highlighted the interest of simulation in data augmentation for deep learning [17,18]. Urban scenes are created using game engines such as "Unreal Engine". These simulated virtual city models increase the amount of training data. Unfortunately, there are few studies on the segmentation of 3D city instances in the literature.

Founded in 2004, Xeos Imagerie Inc. specializes in the acquisition and processing of high-resolution digital aerial photographs as well as LiDAR topographic surveys [19]. Each year, the company carries out over 75,000 km$^2$ of image acquisitions for applications in urban planning, forestry, pipelines, wind farms, mines, erosion zones, etc. Xeos has recently created a program called "Xeos 3D Cities", capable of delivering high-quality 3D models of entire cities. This program aims to deliver data extracted from the 3D model directly to customers. To automate its processing chains, Xeos is embarking on the segmentation of 3D city instances using deep learning. The results of this segmentation will enable the company to produce new layers of information.

To train a 3D city segmentation model with broad applicability across datasets, a substantial volume of annotated data is essential. The training data encompass both actual data representing Quebec City, Canada, and synthetic data generated from diverse platforms like the 3D engine "Unreal Engine" and Evermotion. PicassoNet-II, a semantic segmentation architecture tailored for textured meshes, was developed nearly two years ago [15]. Initially, the semantic segmentation of 3D cities was executed using this model. Subsequently, a context analysis based on Markov fields was incorporated into the algorithm for contextual feature assessment post-semantic segmentation. The algorithm was adapted to iterated conditional modes (ICMs) [20], employed to minimize the energy defined by a random Markov field. Finally, a cluster-based analysis was applied to the semantic segmentation outcomes to extract buildings. It is crucial to clarify that this phase does not utilize artificial intelligence. It does not entail the training of instance segmentation models; instead, it leverages the semantic segmentation output layer to assess object extraction. Other research works have used similar approaches. Ref. [21] used Mask R-CNN with a KMeans kernel on computed tomography images to improve the segmentation performance of the lung region. Ref. [22] implemented the KMeans clustering algorithm to separate vertebral arches from bodies after fully segmenting the spine using a 3D convolutional neural network. This approach to instance segmentation underscores the need for clustering mechanisms that can handle the variability in instance shapes and sizes inherent in urban landscapes. By incorporating such clustering techniques, the segmentation process can be refined to recognize individual buildings of 3D city models. The joint use of simulated and real data, the adaptation of a Markov algorithm to 3D meshes, and the use of cluster analysis algorithms are complex and innovative tasks in the context of digital twins of 3D cities. The aim of this article is to highlight the potential benefits of using data simulation for building extraction, taking into account not only geometry but also textural information, such as colors in the RGB mode.

## 2. State of the Art

### 2.1. Three-Dimensional Cities

A 3D city is a model that represents the 3D geometry of urban elements [1,23,24]. The 3D geometry of the landscape can be formed by a point cloud or a 3D mesh on which textures can be applied. A mesh is a geometric network composed of segments, facets, and vertices that form a network of triangles to approximate the geometric surface of 3D objects [12,13,15]. A textured mesh is one in which textures have been projected onto the facets. The term "texture" is used in the 3D domain to designate the RGB image projected onto the mesh, which provides information on the color of the mesh facets.

### 2.2. The Contribution of Artificial Intelligence

The rapid development of computer processing power and the exploitation of graphics processing units (GPUs) has enabled the emergence of deep learning, a sub-field of artificial intelligence [25]. It is based on the use of deep artificial neural network (ANN) architectures, simulating the functioning of neurons in the human brain. A neural network is composed of several interconnected layers of neurons [26].

The extension of deep learning to image processing is based on the addition of a convolution layer, a kind of filter bank that extracts properties from the image. This innovation gave rise to convolutional neural networks (CNNs). The accuracy of image segmentation is assessed using intersection over union (IoU), a metric commonly used to measure the effectiveness of a model. IoU is the ratio of correctly detected features (true positives) to the union of true positives with features not detected (false negatives) or detected in error (false positives).

### 2.3. Three-Dimensional Semantic Segmentation

Interest in 3D semantic segmentation has grown in recent years thanks to the successful application of CNNs on images [16,27]. Although the use of CNNs on 2D data has become commonplace, the application of these types of neural networks is not as obvious with 3D data, such as point clouds and meshes. Indeed, the unordered nature of 3D data seems a priori incompatible with image convolution filters. Several mesh segmentation techniques have been developed [11,15,16]. Unlike point clouds, a mesh has a topological structure that provides more precise geometric information, as shapes are represented by continuous polygonal surfaces and not by a cluster of points. This has the advantage of reducing the storage space required compared with point clouds [16,28].

There are various approaches to mesh segmentation. Some methods use convolutions based on local patch operators with arbitrary coordinate systems [11,29]. Other studies have explored methods based on multiview labeling [18,30]. There are also methods using spatial graph convolutions (SCGs) [28,31,32]. Finally, recently, studies have begun to implement CNNs using 3D convolution filters on triangle networks [15,16].

Ref. [15] developed a neural network called PicassoNet-II that enables semantic segmentation to be performed on textured meshes. ScanNet and S3DIS data were used to test this segmentation. PicassoNet-II achieved an average IoU of 69.8% and 68.6% on S3DIS and ScanNet data, respectively, outperforming KPConv. The results are shown in Appendix A and are referred to as Figures A1 and A2.

### 2.4. The Role of Real and Simulated Datasets

Training semantic segmentation models requires the preparation of a lot of extensive, high-quality annotated data, which is a laborious process. For this reason, there are a number of image, point cloud, and mesh datasets in the literature to facilitate model performance comparisons. To facilitate model performance comparisons, various image, point cloud, and mesh datasets are available in the literature. Notable datasets, such as the "Stanford 3D Indoor Scene Dataset" (S3DIS) [14,18], ScanNet [15,33], and Matterport3D [11,34], have played pivotal roles in indoor scene analyses. These datasets, rich in point clouds and meshes, are instrumental for benchmarking. Recently introduced outdoor urban datasets

like "Hessigheim 3D" [35] and "SUM-Helsinki" [28], derived from aerial LiDAR scans and imagery, broaden the scope of research.

Simultaneously, researchers are exploring the potential of "3D engines" to simulate urban environments and work with digital twins [36–39]. These engines enable users to generate a large amount of data within a virtual universe, offering control over various factors. Among other things, these data can be annotated automatically by algorithms that are more efficient than manual annotation [40]. The Blender and Unreal Engine software packages are particularly interesting for research, as it is possible to develop one's own tools [41,42]. Unreal Engine, developed by the company "Epic Games Inc.", enables users to create virtual worlds [43], and its platform includes a web store with realistic 3D models. Blender, a free software for creating and manipulating 3D models, enhances accessibility [44].

In the context of the growing need for abundant annotated data in deep learning models, synthetic data emerge as a pivotal solution [45]. Synthetic data offer advantages such as being easy to generate and having error-free attributes, pre-annotated capabilities, and ethical considerations. The ongoing progress in data synthesis and domain adaptation techniques is closing the statistical gap between synthetic and real data. Beyond sustaining the deep learning revolution, synthetic data hold promise for a next generation of models that understand the physical world, facilitating continual, multimodal, and interactive learning.

## 3. Materials and Methods

### 3.1. Study Area

This project's study site is Quebec City (Figure 1). It is the capital of the province of Quebec, and it is its second largest city. The enlargement window corresponds to the specific study area selected, from which the actual data were extracted. This procedure is described in detail in Section 3.2.

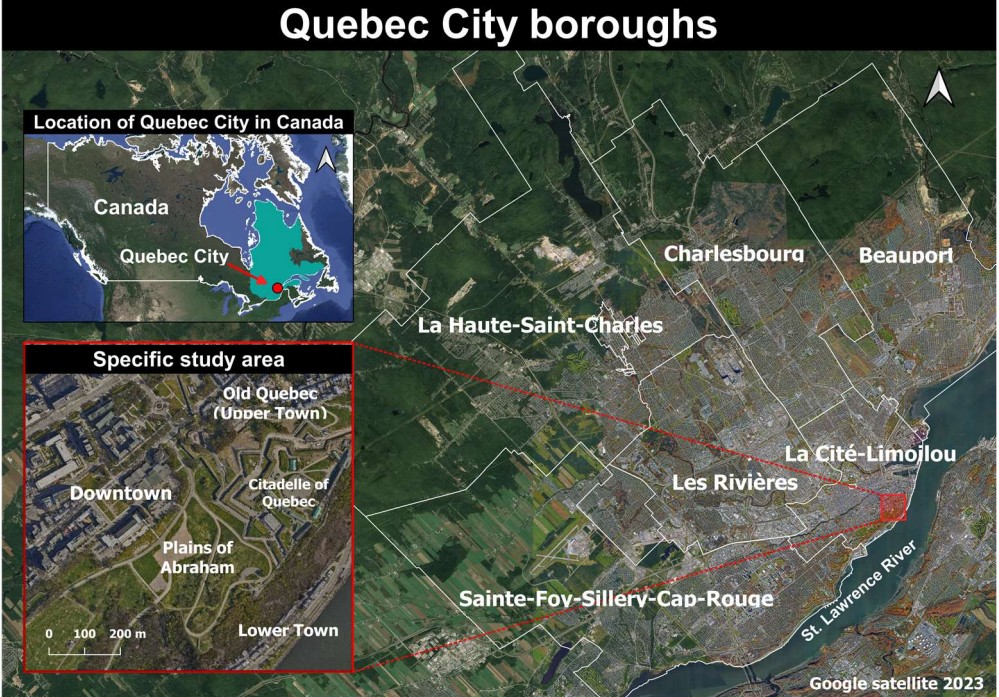

**Figure 1.** Study site (Quebec City).

### 3.2. Preparation of the Real Dataset

3.2.1. Inventory of Real Datasets

The authentic data are derived from the comprehensive Xeos 3D model that encompasses the entirety of Quebec City, organized into 1 km by 1 km tiles [46]. The georeferencing accuracy of this model is finely tuned to 10 cm. The construction of the 3D model employed a hybrid methodology, combining 5 cm resolution aerial photography and LiDAR data. The specific tile chosen for analysis covers segments of the Saint-Jean-Baptiste and Old Quebec districts, encompassing notable landmarks such as the Quebec National Assembly, the Plains of Abraham, and the Citadel of Quebec. To facilitate processing, this larger tile was subdivided into smaller 110 m by 110 m tiles. Certain tiles, which had an excessive number of vertices, were subdivided further. In total, 114 tiles were generated from the larger dataset for analysis. The enlarged study site is shown in Figure 2. This area has the advantage of being very diverse in terms of its landscape. Among other things, it contains steep cliffs, skyscrapers, wooded areas, urban sectors of varying density, and so on. Tiles of different sizes and themes were chosen to provide a training dataset representative of the territory. Examples are available in Appendix B and are referred to as Figure A3.

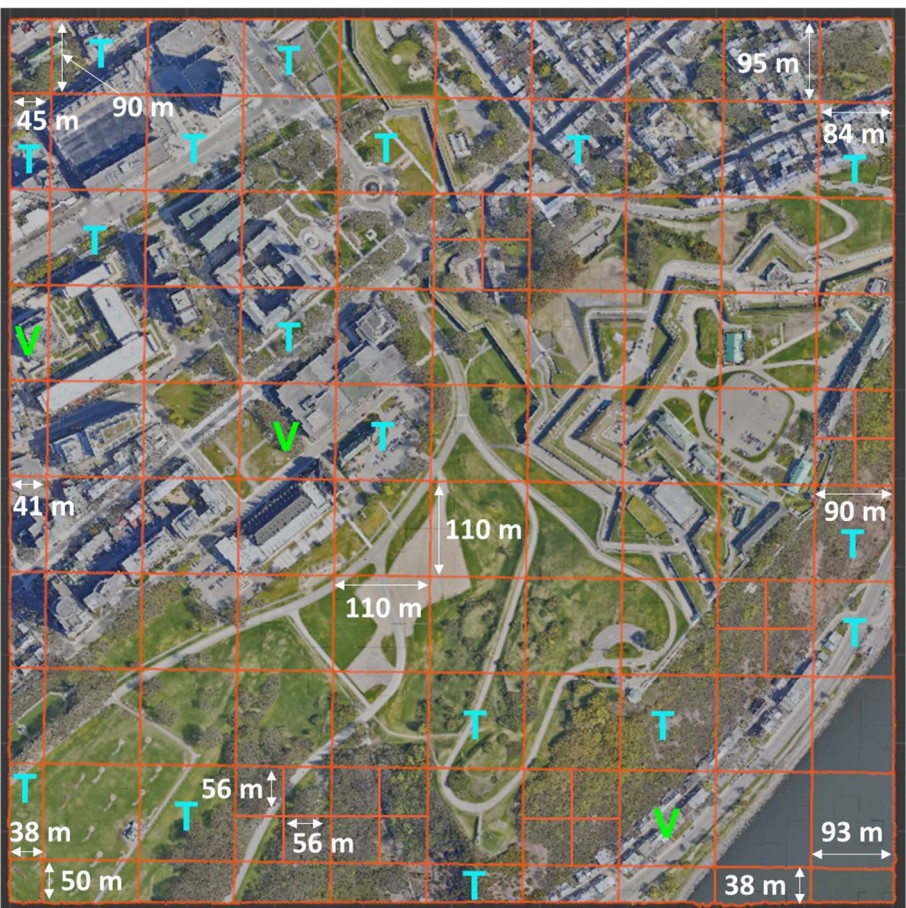

**Figure 2.** Distribution of Xeos 3D model tiles in the training dataset. The letters "T" and "V" stand for "training" and "validation", respectively.

The first semantic segmentation tests on urban scenes were carried out using the SUM-Helsinki dataset. The dataset consists of 64 annotated tiles, with each tile encompassing an area of 250 square meters. In total, these tiles cover approximately 4 square kilometers of the city of Helsinki, Finland. Created in 2017, the dataset was generated from aerial LiDAR scans and aerial imagery. The annotations within the dataset are categorized into six classes: terrain, vegetation, building, water, car, and boat [28].

### 3.2.2. Annotation of Xeos Three-Dimensional Model Tiles

The objects were divided into 5 semantic classes: terrain, vegetation, building, unclassified, and high_urban. The high_urban class in the annotation represents tall, slender, vertical structures such as poles, lampposts, cranes, and similar elements. The unclassified class includes various objects like fences, retaining walls, cars, hydrants, ground cavities, etc. The annotation of the Xeos 3D model was conducted using the "Urban Mesh Annotation Tool" application, developed by [28] specifically for annotating 3D meshes of urban scenes. Notably, the SUM-Helsinki dataset has undergone complete annotation within this application. The annotation process is semi-automatic, and the application includes tools for slicing the 3D mesh into planar segments. Figure 3 illustrates the general procedure for annotating 3D tiles.

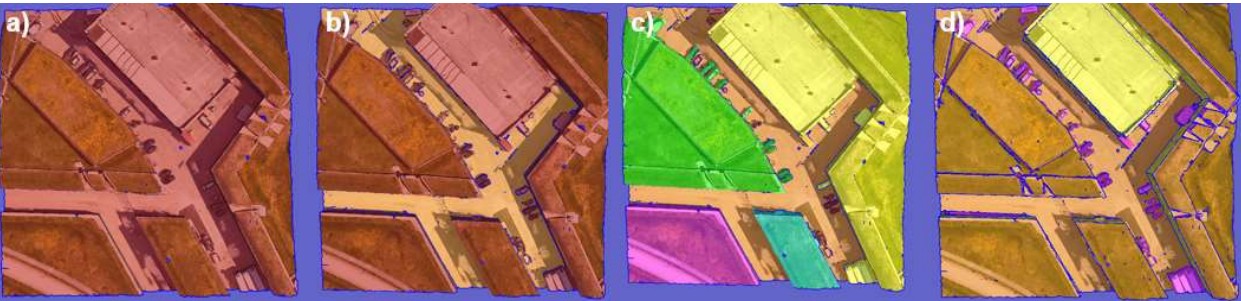

**Figure 3.** Annotation procedure followed using the Urban Mesh Annotation Tool: (**a**) a new mesh is imported; (**b**) the main plane is extracted and annotated; (**c**) the secondary surfaces have been extracted; (**d**) the objects have been divided and annotated into their respective classes.

First, the main plane, which generally corresponds to the ground, is extracted, naturally isolating groups of objects above ground level, such as buildings and trees. The next step is to continue separating the surfaces until the desired objects can be extracted and annotated into their respective classes. Once all objects have been correctly separated, all that remains is to manually correct any incorrectly annotated triangles. The annotation process takes between 1.5 and 4.5 depending on the density of objects in the tile. The annotation has been performed on a laboratory computer with the following specs: NVIDIA RTX 3080 (8 GB of VRAM), Intel Core i7-11800H @ 2.30GHz, and 64 GB of RAM. Examples of annotated tiles have been compiled in Appendix C and are referred to as Figure A4.

With the completion of this phase, only one task remained—generating the five customary files, a process that will be further described in Section 3.4. The 114 real tiles were then ready to be used by PicassoNet-II.

### 3.3. Preparation of the Simulated Dataset

3.3.1. Inventory of Simulated Datasets

The simulated training data comprise 3D models of buildings primarily obtained from two specialized platforms and websites dedicated to the sale of such models, namely "Evermotion" and "Epic Games Market." Evermotion, a Poland-based company renowned for creating realistic 3D models of scenes [47], contributed a total of six city models to the project. However, only the three most straightforward models to prepare were utilized for this specific project (see Figure 4). Additionally, an independent artist's model named KC15 was also purchased and included in the training data.

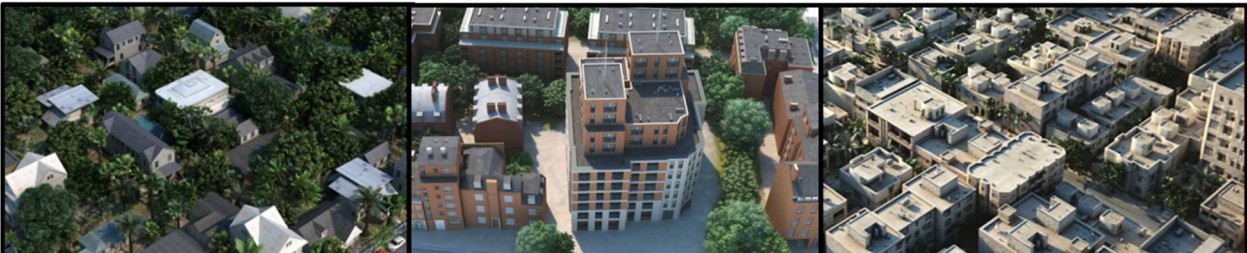

**Figure 4.** Extracts from the three Evermotion models selected. From left to right: AM131-City001, AM131-City003, and AM133-City002. Taken and modified from Evermotion [48,49].

The Epic Games Market platform hosts a wide array of realistic 3D models encompassing structures and cityscapes. These models are designed for direct utilization within the Unreal Engine and can also be exported to other software applications. Modules purchased from the store total around 150 different buildings, as well as cars, lampposts, power poles, and more.

The objects were divided into 5 semantic classes: terrain, vegetation, building, car, and high_urban. During the content analysis of the models, it was found that the diversity of objects was not constant for each model. For example, model AM131-City003 contained only trees, buildings, and soil, while model AM131-City001 contained utility poles, cars, fences, and even swimming pools. Some objects were therefore added and removed as required. The objects added came mainly from other Evermotion models, but some of the cars and utility poles came directly from Unreal Engine. These objects have been compiled in Appendix D and are referred to as Figure A5. In particular, it was possible to create variations of certain objects by changing the color and orientation of the semi-trailer trucks, for example. It was also necessary to apply several corrections to the geometry and textures of the models. For example, all interior structures of the buildings had to be removed, and some erroneous texture files had to be modified. These operations were carried out using Blender software, version 3.6. Each 3D tile is made up of 5 layers of data representing the 5 semantic classes. Dividing the data into distinct layers was essential for the automatic annotation of the unified mesh. Figure 5 shows an extract of completed 3D tiles.

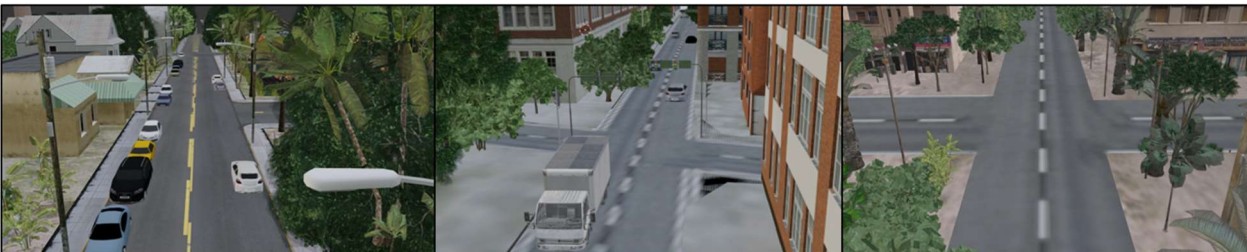

**Figure 5.** Extract from completed Evermotion mock-ups. From left to right: AM131-City001, AM131-City003, and AM133-City002.

3.3.2. Deformation and Labeling of Simulated Tiles

The simulated data exhibit precise geometry and accurate textures, but the mesh lacks unity, meaning there is no connectivity between objects. This presents a challenge, as PicassoNet relies on this connectivity to learn object characteristics. In contrast, real data comprise unified 3D meshes that have undergone deformation through a 3D reconstruction process. Consequently, it becomes essential to replicate the same types of deformation on simulated data to ensure compatibility with real data. To achieve this, Blender's "Remesh" and "Union" tools were essential. The "Remesh" tool transforms an input mesh into a simplified mesh that attempts to respect the shape of the original mesh as closely as possible. The "Union" tool, on the other hand, is used to merge 3D meshes together, creating connectivity between all objects [44].

The operations performed in Blender are not sufficient to make 3D tiles usable. Indeed, as shown in Figure 6b, Blender not only generates too many vertices, but the size and distribution of the triangles are far too different from one class to another. The GraphiteThree application, version 3-1.8.0., was used to solve both of these problems. This application, developed by [50], specializes in 3D mesh processing. This application has a "Remesh" tool that allows both the density and size of triangles in a mesh to be controlled. Each simulated tile was processed in GraphiteThree to obtain a number of vertices below 500,000 and distribute the triangles more evenly throughout the mesh. Figure 6 illustrates the three stages of mesh deformation: original mesh, deformed mesh in Blender, and deformed mesh in GraphiteThree.

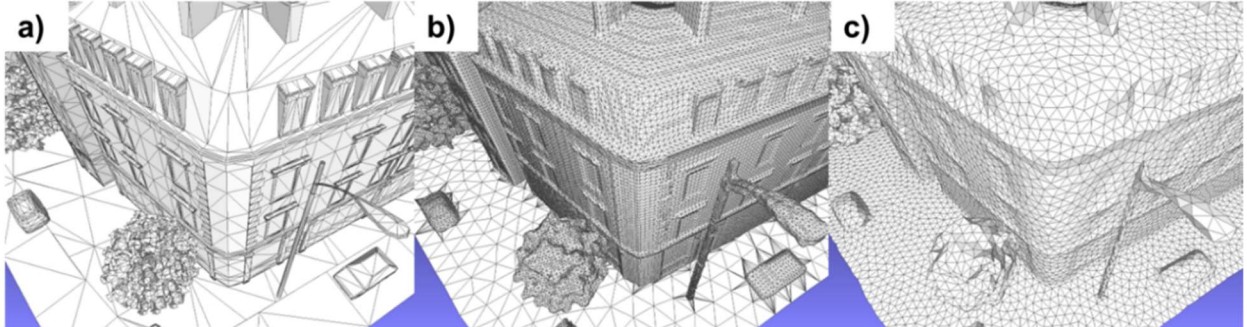

**Figure 6.** State of the 3D mesh at each deformation stage: (**a**) original mesh, (**b**) mesh processed in Blender, and (**c**) mesh processed in GraphiteThree.

After processing the tiles in GraphiteThree, all that remained was to apply the original mesh texture to the unified mesh and annotate the triangles. The whole process was automated with a Python script. The texture of the original mesh (Figure 7a) was applied to the vertices of the unified mesh (Figure 7b) by superimposing the data. Because it is the vertices that receive the color, the textures are of poorer quality than originally. The triangles were automatically annotated by matching the closest triangles by superimposing the 5 layers on the unified mesh (Figure 7c). Finally, all that remained was to generate the 5 files required as input information for PicassoNet. Further details about each of these files are described in the subsequent paragraph. The 29 simulated tiles were then ready to be used by PicassoNet. Detailed views of the 3D models are available in Appendix E and are referred to as Figures A6–A8.

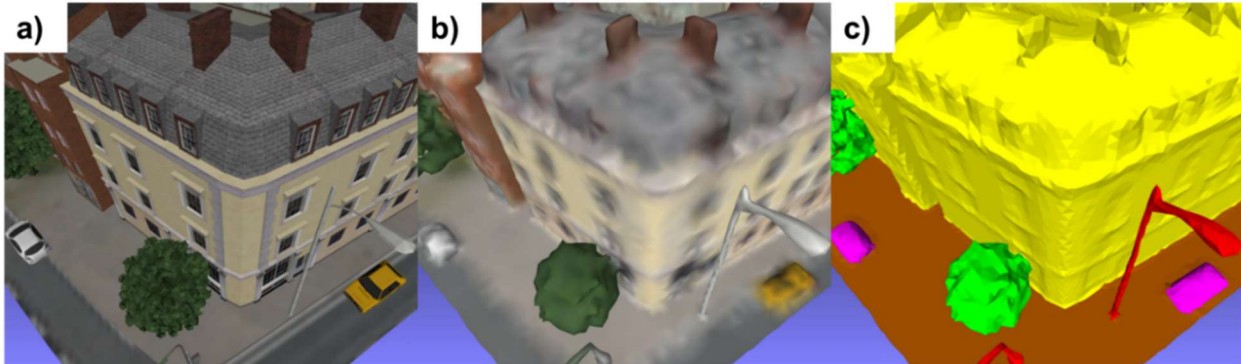

**Figure 7.** Comparison between the original textured mesh and the final result: (**a**) original mesh, (**b**) final textured mesh, and (**c**) final annotated mesh.

### 3.4. PicassoNet-II

The PicassoNet-II neural network [15] was selected for the segmentation of textured 3D meshes, primarily due to its recent development and availability on the web. This

network follows an encoder–decoder architecture and incorporates three key convolutions directly affecting the meshes: "facet2vertex", "vertex2facet", and "facet2facet".

To effectively operate on 3D meshes, PicassoNet requires information to be distributed across five distinct files: the textured 3D mesh, the class number assigned to each vertex, the barycentric coefficients of the vertices, the textured point cloud, and the count of points within each triangle. The textured 3D mesh encompasses vertex coordinates, vertex colors, and facets. Barycentric coefficients represent any coordinate within a triangle as a scalar (ranging from 0 to 1), and the corresponding file includes the barycentric coordinates for each vertex. The textured point cloud includes the coordinates of all vertices along with their color, as well as other sampled points within the triangles.

For the automated generation of these five files for any textured 3D mesh, a Python script utilizing the PyMeshLab [51–53] and Trimesh [54] libraries was designed. PicassoNet's main hyperparameters include the learning rate, the number of vertices to be retained during decimation, and the batch size. As part of this project, PicassoNet was installed on Compute Canada's Graham cluster to leverage available GPUs for training purposes [55]. In the initial testing phase during the initialization of the neural network, it was observed that PicassoNet's processing capacity was restricted, being able to handle a maximum of 500,000 vertices allowed in a batch when utilizing an NVIDIA P100 Pascal GPU.

### *3.5. Contextual Analysis Based on Markov Fields*

3.5.1. Basic Principles

In computer vision and 3D modeling, the semantic segmentation of 3D meshes remains a major challenge due to the complexity and variability of real scenes. This is why Markov field-based contextual analysis is integrated into the methodology, more specifically the iterative conditional modes (ICM) algorithm, to improve results following the semantic segmentation of 3D meshes.

Markov fields (MRFs) are a class of probabilistic models used to capture contextual relationships between neighboring features in various image processing and computer vision applications [56]. They offer a powerful mathematical framework for modeling spatial dependency between discrete or continuous random variables. MRFs are particularly useful for solving image segmentation, denoising, and reconstruction problems, where contextual consistency is crucial [57,58]. Mathematically, an MRF can be defined as follows:

Let $X = \{X^1, X^2, \ldots, X^n\}$ be a set of random variables corresponding to features of interest, such as pixels in an image. Markov fields model the dependencies between these variables using interaction potentials. The total energy of an MRF is given by:

$$E(X) = \sum_i \psi_i(X_i) + \sum_{i,j} \psi_{i,j}(X_i, X_j) \tag{1}$$

where $\psi_i(X_i)$ is an individual potential associated with the random variable $X_i$, and $\psi_{i,j}(X_i, X_j)$ is an interaction potential between the variables $X_i$ and $X_j$. The aim is to find an $X$ labeling configuration that minimizes this total energy, reflecting optimal segmentation or assignment of variables. MRFs are often solved using optimization methods such as the ICM algorithm mentioned earlier to find the optimal labeling that respects both local constraints and global interactions.

3.5.2. Adapted Algorithm for ICM

The ICM algorithm is an optimization method widely used to solve energy minimization problems on Markov fields (MRFs) [20]. The main objective of ICM is to find the labeling configuration that minimizes the total energy of the MRF, taking into account local interactions and contextual constraints, i.e., its direct neighborhood. The ICM algorithm works iteratively by updating the labels of the MRF's random variables, optimizing them one by one while keeping the labels of neighboring variables constant. The process is

repeated until convergence is reached, i.e., label changes no longer result in a significant improvement in total energy.

In this project, local energy is calculated as follows for a vertex, its neighborhood, and a class (*c*):

$$E_{ICM}(c) = \beta \cdot \left( \sum_{j \in neighbors(i)} \|[y_j \neq c] - \sum_{j \in neighbors(i)} \|[y_j = c] \right) - \log(P(y_i = c)) \quad (2)$$

where $\beta$ is the beta parameter (varying from 0 to 1) that governs the influence of neighboring vertices, $j$ is a particular neighboring vertex, $i$ is the target vertex, $neighbors(i)$ denotes the set of neighboring vertices selected for vertex $i$. $y_j$ is the predicted class for vertex $i$, and $c$ is a specific class label; $\|[\cdot]$ is the indicator function that evaluates to 1 when the condition inside is true, and it is 0 otherwise. $P(y_i = c)$ represents the predicted probability that vertex $i$ belongs to class $c$. In this formula, the first term captures the compatibility between class $c$ and the classes of neighboring vertices. The second term incorporates the probability of assigning class $c$ to the current vertex based on the probabilities predicted by the model (e.g., the softmax function). The ICM algorithm aims to determine the lowest energy by iteratively evaluating it for various class assignments, and it will thus determine the most likely class. This results in a more consistent and probable segmentation.

### 3.5.3. Adaptation to Three-Dimensional Meshes

The algorithm for selecting the neighbors of the target vertex has been adapted to exploit the connectivity of the 3D mesh. An additional parameter "αl" was incorporated into the algorithm, allowing the neighborhood level to be specified for selection, as visualized in Figure 8a. In addition, to ensure consistency with the target vertex, the angular defect was calculated for each of the relevant vertices. In particular, only those neighbors whose angular defects lie within the interquartile range (IQR) were retained. This approach ensures the avoidance of neighbors whose geometries could differ significantly from that of the target vertex. As illustrated in Figure 8b, the two vertices at the foot of a building were correctly excluded from the selection, their value being outside the upper limits of the IQR (0.000067 in this example).

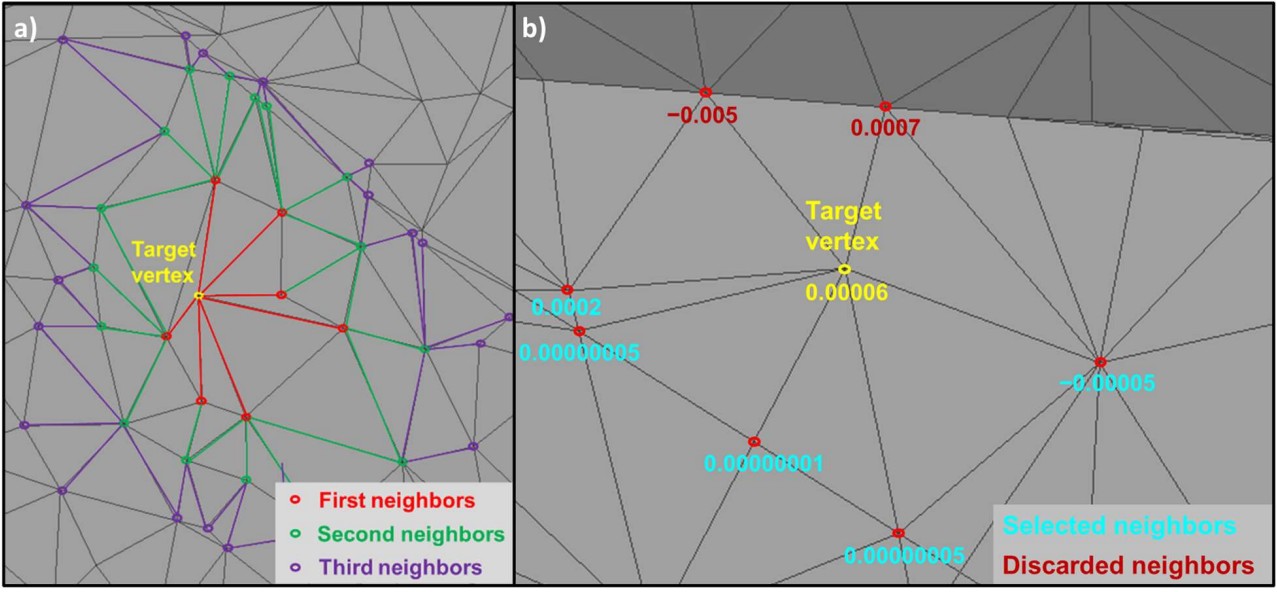

**Figure 8.** Procedure for selecting neighboring vertices according to their neighborhood level (**a**) and angular defect (**b**).

To limit the simultaneous processing of all vertices, a final measure was introduced. An additional parameter named "p" has been incorporated to retain exclusively those

vertices for which the difference between the two maximum softmax indices is less than or equal to a given value. This approach was designed to avoid processing vertices for which there is no doubt about the predicted class.

### 3.6. Cluster Analysis Algorithms

Cluster analysis is a fundamental method in unsupervised machine learning. It aims to group similar data into sets called clusters. The scikit-learn (sklearn) library offers a range of cluster analysis algorithms for exploring latent structures in data. For this project, three of these algorithms are explored: KMeans, spectral clustering, and DBSCAN [59].

### 3.6.1. KMeans

The KMeans algorithm divides the data into $\kappa$ clusters by minimizing the sum of the squared distances between the points and centroids of their respective clusters. KMeans relies on the random initialization of centroids, followed by alternating steps of assigning points to clusters and updating centroids. This iterative approach generally converges on a stable solution. KMeans is suitable for datasets where clusters have convex shapes and similar sizes. The main parameters of the algorithm are:

- n_clusters: The number of clusters to be formed;
- init: The centroid initialization method;
- n_init: The number of different initializations of the KMeans algorithm to try. The final result is the one that minimizes the sum of squared distances;
- max_iter: Maximum number of iterations for each KMeans initialization.

### 3.6.2. Spectral Clustering

The spectral clustering algorithm draws on the concepts of graph theory and spectral decomposition. It transforms data into a graphical representation and exploits the graph's spectral properties to perform partitioning. Using an eigenvalue-based approach, spectral clustering identifies clusters that are not necessarily convex or of equal size. This makes it a relevant choice for data with more complex, non-linear structures. The main parameters of the algorithm are:

- n_clusters: The number of clusters to be formed;
- affinity: The similarity measure used to construct the affinity matrix;
- assign_labels: The method for assigning labels to clusters.

### 3.6.3. DBSCAN

DBSCAN (Density-Based Spatial Clustering of Applications with Noise) is a robust algorithm for detecting clusters of varying density in noisy datasets. Rather than fixing a number of clusters $\kappa$ in advance, DBSCAN automatically identifies high-density regions based on a radius $\varepsilon$ and minimum number of neighbors. Points that do not meet these criteria are considered noise. DBSCAN is particularly useful for processing data where clusters may have complex shapes and uneven densities. The main parameters of the algorithm are:

- eps: The radius around a point to define its neighborhood. This is a crucial parameter for DBSCAN, as it defines the maximum distance between two points for them to be considered neighbors;
- min_samples: The minimum number of points required for a point to be considered a nucleus (central point of a cluster);
- metric: The distance metric used to calculate distances between points;
- algorithm: The algorithm used to calculate neighbors.

In short, the scikit-learn library offers a diverse range of cluster analysis algorithms, including KMeans, spectral clustering, and DBSCAN. Each algorithm has its own advantages and is adapted to specific types of data and cluster structures. The choice of algorithm depends on the characteristics of the dataset and the objectives of the cluster analysis.

## 4. Results

### 4.1. Semantic Segmentation Using PicassoNet-II

#### 4.1.1. Results of Training with the SUM-Helsinki Dataset

The 64 tiles in the SUM-Helsinki dataset were distributed as follows during the training phase: 40 for training, 12 for validation, and 12 for inference (see Figure A9 in Appendix F). During training on the SUM-Helsinki dataset (as illustrated in Figure 9), we can see that the IoUs for the "building" and "vegetation" classes are already very high (>90%) and remain constant throughout training. The "terrain" class also remains stable at around 80%. These three classes are the best represented in the dataset. The results for the "car" and "water" classes show a slight improvement from epoch 1 to epoch 2, with IoUs ranging from around 64% to 74%. The "boat" class showed the greatest improvement, with an IoU rising from around 45% to 85%. Each epoch took around 7 h. It is essential to emphasize that, while epoch 13 produces the best results, there is a possibility of achieving even better results with extended training.

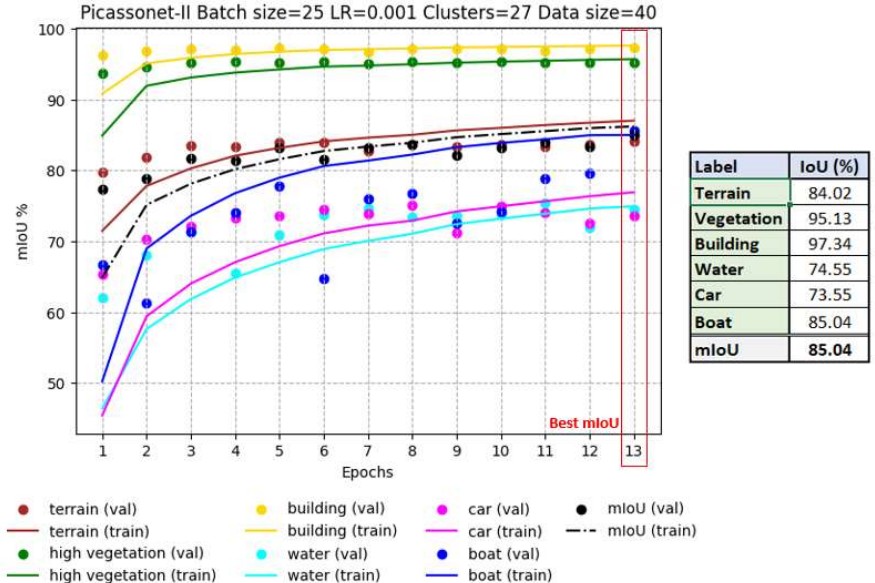

| Label | IoU (%) |
|-------|---------|
| Terrain | 84.02 |
| Vegetation | 95.13 |
| Building | 97.34 |
| Water | 74.55 |
| Car | 73.55 |
| Boat | 85.04 |
| mIoU | **85.04** |

**Figure 9.** Progression of validation results by class at each epoch during SUM-Helsinki tile training (**left**). IoUs of the best result (**right**).

As can be seen in Figure 10, it is difficult to distinguish ground truth from predictions. These results confirm that PicassoNet works and looks promising for 3D city segmentation.

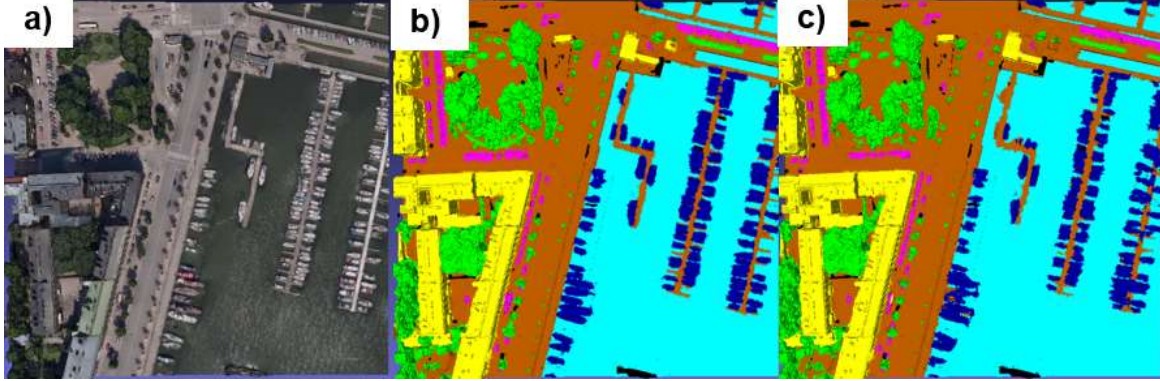

**Figure 10.** Inference results on a test tile. Comparison between actual colors (**a**), ground truth (**b**), and predictions (**c**) for a tile in the SUM-Helsinki test dataset.

4.1.2. Training Results with Simulated Data Only

Evermotion models are made up of 29 3D tiles. During training, the data were divided as follows: 21 tiles for training, 7 tiles for validation, and a single tile (KC15) for inference. The learning rate for all training runs was set at 0.001 and decayed according to an exponential decay formula. It is essential to note that the number of epochs performed during the different training runs is not uniform. This variation can be explained by significant fluctuations in the execution time per epoch, directly linked to the size of the dataset. It should also be pointed out that these variations are also influenced by the processing time limit imposed by Compute Canada.

The most notable results were recorded in epoch 6, with an average IoU reaching 98.72%, including 99.50% for buildings (see Figure 11). In particular, the validation IoU for the "car" class increased significantly from 86% to 97% between epoch 1 and epoch 6, while the other classes had already reached a plateau by epoch 3. These observations suggest that the model easily captured the characteristics of the objects present in the simulated tiles. Detailed graphs of the evolution of loss and accuracy for all training sessions are available in Appendix G and are referred to as Figures A10–A13.

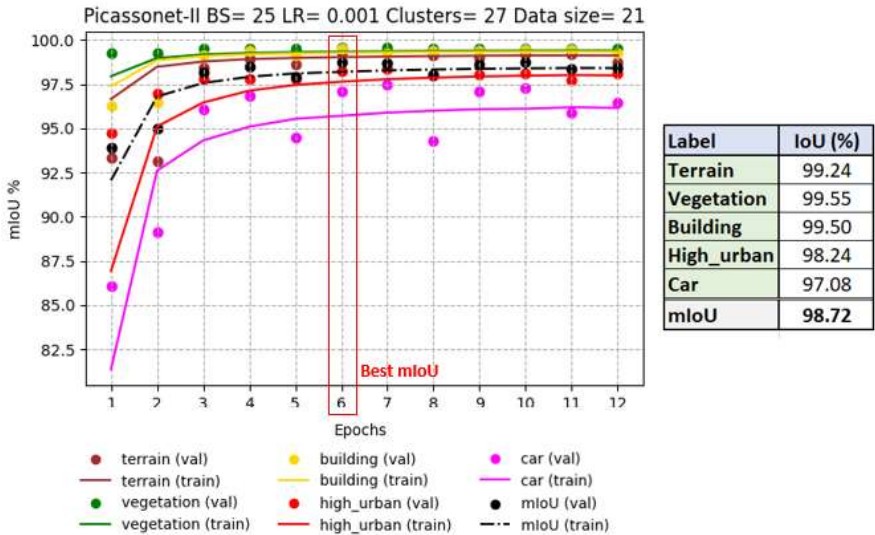

**Figure 11.** IoU progression for each class during PicassoNet-II training on simulated data (**left**). IoUs of the best result (**right**).

To assess the model's generalizability to other datasets, an inference was conducted on the KC15 tile (simulated dataset) and a Xeos tile (real dataset).

For the inference on the KC15 tile, the outcomes were reasonably definitive, showing an average IoU of 92%. Notably, the primary errors were associated with features that were either absent or under-represented in the training dataset. These features included hedges, low vegetation, cavities, and certain street lamps and traffic lights with a horizontal component (Figure 12).

As for the inference on Xeos tiles, the results were relatively disappointing, with an average IoU of just 48% for the three validation tiles. Figure 13 shows a tile depicting the facade of the Quebec National Assembly, where significant confusion between the "building" and "high_urban" classes can be clearly seen. These errors are visually inconsistent, as they occur on many parts of building facades. In addition, many artifacts appear at ground level. However, after processing the tile with GraphiteThree, the inference results are significantly improved. The confusion between the two classes has completely disappeared, as have most of the ground-level artifacts. These results highlight the sensitivity of the model to the method of creating the 3D mesh itself.

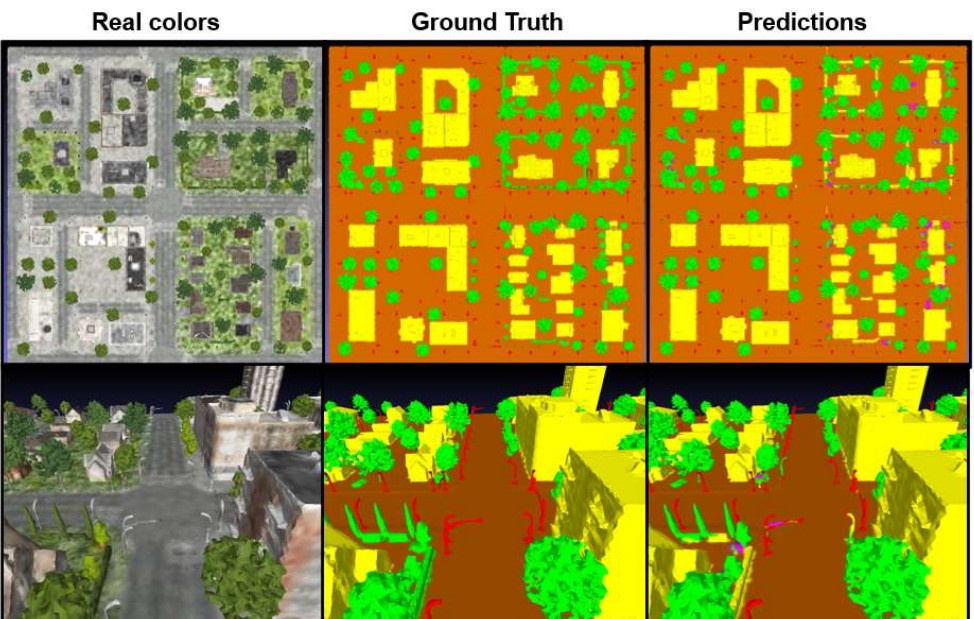

**Figure 12.** Inference results for the KC15 tile under different views.

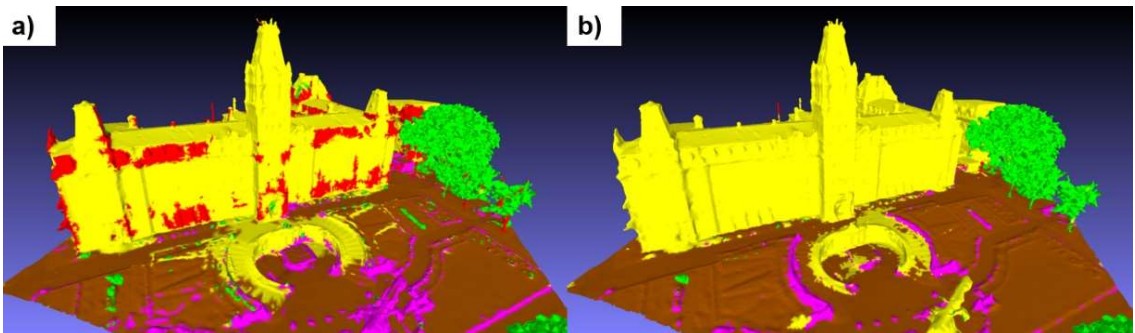

**Figure 13.** Comparison of inference results for a Xeos tile with its geometry intact (**a**) and the same tile after being modified in GraphiteThree (**b**).

4.1.3. Training Results on Real Data Only

Tiles for the Xeos 3D model were distributed as illustrated previously in Figure 2, where 17 tiles were used for training and 3 were reserved for validation. The results obtained are shown in Figure 14, highlighting optimal performance at epoch 12, with an average IoU of 79.87%. This result is particularly notable for the "building" class, where an IoU of 95.68% was achieved.

Interestingly, the IoU in validation remains stable for most classes, with the exception of "high_urban" and "unclassified", which show marked variations from one epoch to the next. It seems that the model suffers from overlearning for the "high_urban" class, indicating a tendency to memorize training data rather than generalize to new data. This observation is not surprising, as these classes are the least represented in the dataset. The "unclassified" class, which groups together various objects that do not belong to any of the four main categories, also presents learning difficulties, which is understandable given its heterogeneous nature. However, it is encouraging to note a slight improvement over time, with the validation IoU rising from around 55% to around 62%.

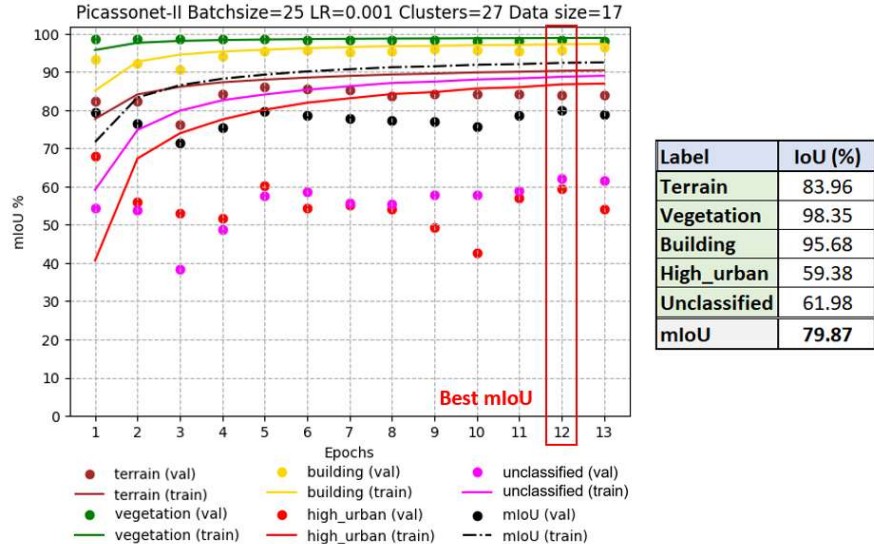

**Figure 14.** Progression of the IoU of each class during PicassoNet training on real data (**left**). IoUs of the best result (**right**).

It is noteworthy that the frequent merging of poles with vegetation during the generation of the Xeos 3D model introduced ambiguity regarding their geometry and texture. While this confusion was more evident in the "high_urban" class, it also had repercussions for other classes. The interplay of these factors, coupled with the limited representation, likely elucidates the difficulties encountered by the model in learning features, especially within the "high_urban" class.

4.1.4. Training Results with Mixed Data (Real and Simulated)

Simulated and real data were combined to form a mixed dataset. This combination includes 18 simulated tiles in addition to 10 real tiles, resulting in a training set of 28 tiles, with the same 3 real tiles being dedicated to validation. Note that these three validation tiles are the same as those used in the previous training phase. The best performance was observed in epoch 16, with an average IoU of 78.04%, including 93.21% for the "building" class (see Figure 15).

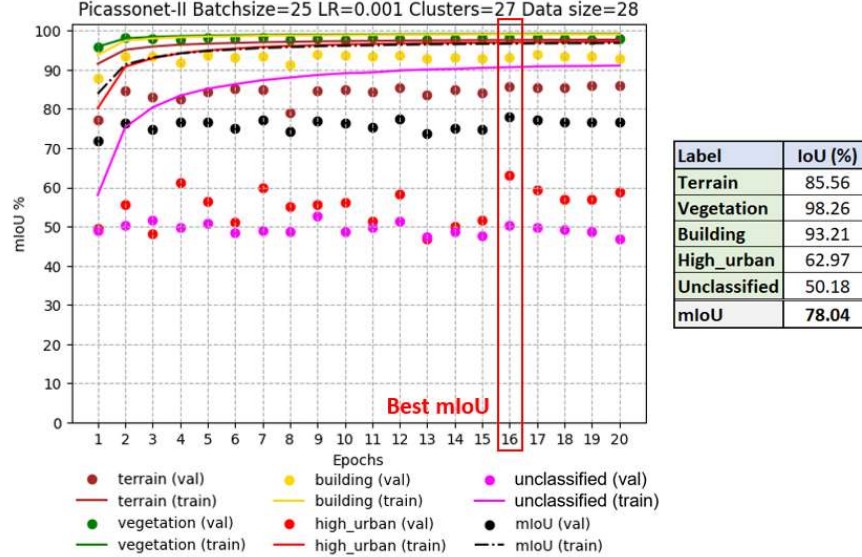

**Figure 15.** Progression of the IoU of each class during PicassoNet training on mixed data (**left**). IoUs of the best result (**right**).

On the one hand, similar to training based exclusively on real data, the validation IoU for the "high_urban" class showed significant fluctuations. However, overall, a clear improvement can be observed, from around 50% at the start to around 60% at the end of training. On the other hand, for the "unclassified" class, results are now much more stable, with the IoU remaining at around 50%. However, in contrast to the previous training, a significant improvement is not observed. This trend could be attributed to the predominant influence of the simulated data, mainly composed of vehicles, whereas the real data of the "unclassified" class encompass a diversity of objects of different natures. The other three classes remain relatively stable in their performance.

### 4.1.5. Results of Transfer Learning on Real Data (Fine-Tuning)

The training phase involves refining the model that was initially trained solely on simulated data. This fine-tuning stage utilizes real data exclusively, aiming to enhance the model's performance. The decision was made to recommence training from epoch 7 onward, based on the recognition that the optimal results were achieved during epoch 6. In epoch 7, the learning rate is reduced to $5 \times 10^{-5}$, a common practice when fine-tuning models to avoid the loss of features already acquired. The data distribution remains the same as in the previous training, where only real data were used.

As shown in Figure 16, optimal results are observed at epoch 10. The average IoU reaches 80.37%, with 96.37% for the "buildings" class. A slight improvement in results appears once again for the "high_urban" class. However, the validation IoU for the "unclassified" class continues to fluctuate throughout the training. In addition, the model shows signs of overfitting from epoch 11 to 14 for these two classes. Overall, this is the model with the best performance in terms of average IoU in validation among the three models evaluated.

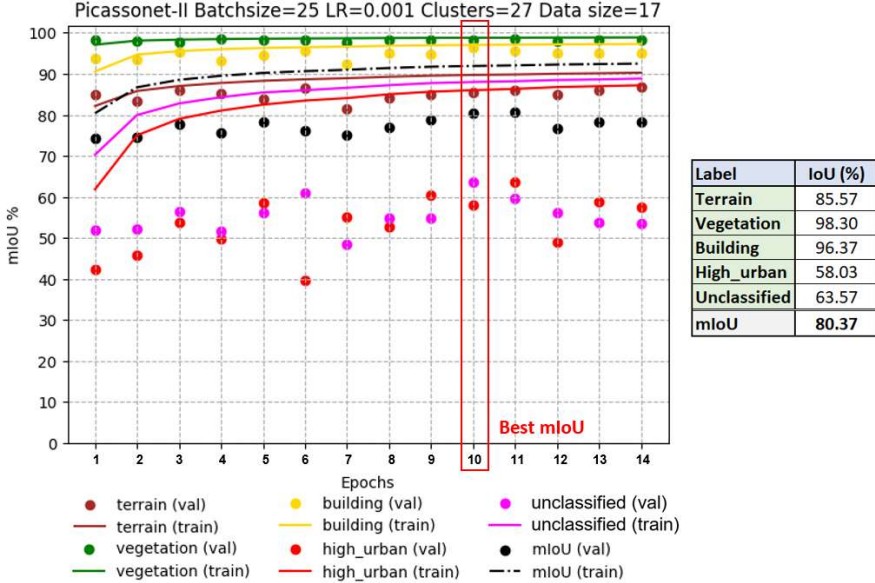

**Figure 16.** Progression of the IoU of each class during PicassoNet training on real data by fine-tuning (**left**). IoUs of the best result (**right**).

In order to assess the impact of the size of the training data set on the results, two other models were tested. In this configuration, the number of training tiles was reduced from 17 to 8, and one of the validation tiles was replaced by another containing fewer posts. Looking at the left of Figure 17, which illustrates training exclusively on real data, it is apparent that there is overlearning for the "unclassified" and "high_urban" classes. In particular, the latter performs very poorly, almost approaching an IoU of 0%. On the right-hand side of Figure 17, we observe the training of real data with a fine-tuning phase. It is immediately apparent that the validation IoU for the "high_urban" class reaches around

50% in the first epoch. It then falls to between 30% and 40% before rising again to between 40% and 50%. The model trained with the fine-tuning phase performs significantly better for the "high_urban" class.

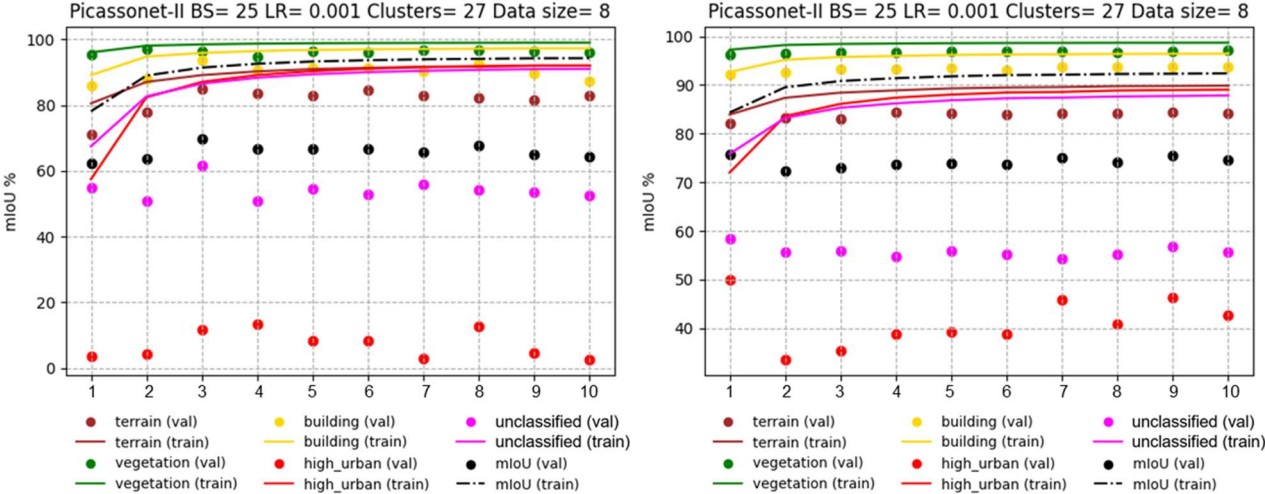

**Figure 17.** Real data training only with reduced data set (**left**) and real data training with fine-tuning (**right**).

Table 1 summarizes the IoU results of the three generated models. Finally, the model trained by transfer learning obtained the best average IoU as well as the best IoU for the "building" class. This suggests that the use of simulated data had a positive impact on training with real data.

**Table 1.** Compilation of the best validation IoUs for the different models generated with PicassoNet-II according to the different datasets used during training.

| Trained Model | IoU (%) | | | | | |
|---|---|---|---|---|---|---|
| | Mean IoU | Terrain | Vegetation | Building | High_urban | Unclassified |
| Simulated data [1] | 98.72 | 99.24 | 99.55 | 99.50 | 98.24 | 97.08 |
| Real data | 79.87 | 83.96 | 98.35 | 95.68 | 59.38 | 61.98 |
| Mixed (real and simulated) | 78.04 | 85.56 | 98.26 | 93.21 | 62.97 | 50.18 |
| Real data (transfer learning) | 80.37 | 85.57 | 98.30 | 96.37 | 58.03 | 63.57 |

[1] The "unclassified" category corresponds to the "car" for this dataset.

The inference results are visually very similar from one model to another. These have been compiled in Appendix H (Figure A14) and are derived from the model trained by transfer learning on the 14 real tiles.

### 4.2. Results of Contextual Analysis Based on Markov Fields

A contextual analysis was carried out to assess the impact of this approach on the semantic segmentation results. Initial tests were carried out on a tile from the Xeos database, used as part of our validation. The mesh of this tile comprises a total of 328,600 vertices. In these tests, the $\beta$ parameter was set to an average value of 0.5. In addition, to maximize variation, the algorithm was repeated five times for each test. A context analysis performance was evaluated for different levels of neighborhood ($\alpha_l$) and probability difference thresholds ($p$), For example, an iteration with $\alpha_l = 1$ and $p = 1$ took around 4 min, while with $\alpha_l = 3$ and $p = 1.0$, it required around 20 min.

The results, shown in Table 2, indicate a quantitative improvement in performance for the "building" class when $\alpha_l < 3$. Overall, however, the results show a decrease in average IoU compared with initial values (before the application of context analysis). By increasing the neighborhood level $\alpha_l$, this reduction in IoUs intensifies.

**Table 2.** IoU results for the different tests carried out with the Markov context analysis algorithm according to different parameters.

| Algorithm Parameters ($\beta = 0.5$) | | IoU (%) | | | | | |
|---|---|---|---|---|---|---|---|
| | | Mean IoU | Terrain | Vegetation | Building | High_urban | Unclassified |
| Default predictions | | 69.17 | 79.85 | 97.56 | 88.87 | 33.60 | 45.99 |
| $\alpha_l = 1$ | $p = 1.0$ | 68.71 | 77.71 | 97.56 | 89.19 | 34.03 | 45.05 |
| | $p = 0.1$ | 68.73 | 77.81 | 97.56 | 89.21 | 34.03 | 45.03 |
| $\alpha_l = 2$ | $p = 1.0$ | 67.37 | 73.69 | 97.37 | 88.98 | 33.26 | 43.56 |
| | $p = 0.1$ | 67.53 | 74.23 | 97.38 | 89.03 | 33.49 | 43.54 |
| $\alpha_l = 3$ | $p = 1.0$ | 66.04 | 69.87 | 97.17 | 88.42 | 33.49 | 41.27 |
| | $p = 0.1$ | 66.34 | 70.95 | 97.19 | 88.63 | 33.49 | 41.43 |

Increasing the neighborhood level has the advantage of more effectively correcting misclassified vertices located within homogeneous zones. However, this approach leads to a significant overlap between several semantic zones, as illustrated in Figure 18. Furthermore, it is important to note that the contextual analysis performs significantly better when applied to only a fraction of the vertices ($p = 0.1$). As shown in Figure 19, this approach succeeds in correcting many isolated vertex clusters and homogenizing semantic boundaries.

For a more in-depth understanding of the impact of the contextual analysis, a more targeted approach was adopted. Instead of calculating the IoU on all vertices simultaneously, the analysis was stratified into two distinct groups: vertices located at the semantic boundary between two classes and vertices located within these boundaries. This strategy enabled us to explore more precisely the algorithm's ability to segment areas where class boundaries are most complex and subject to change. By examining these two groups of vertices separately, it is possible to assess how the contextual analysis contributes to a better delineation of semantic transition zones in relation to interior regions. This detailed analysis offers crucial insights into the performance of context analysis in more delicate segmentation scenarios, where accuracy and consistency are essential.

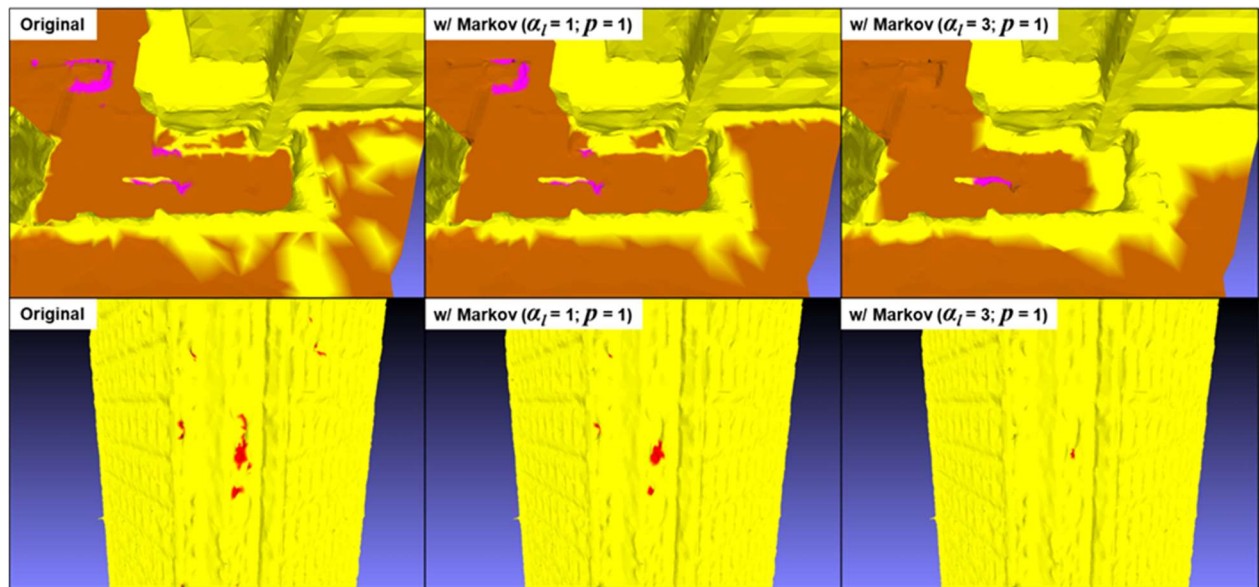

**Figure 18.** Visual impact of Markov context analysis on Xeos 3D tiles.

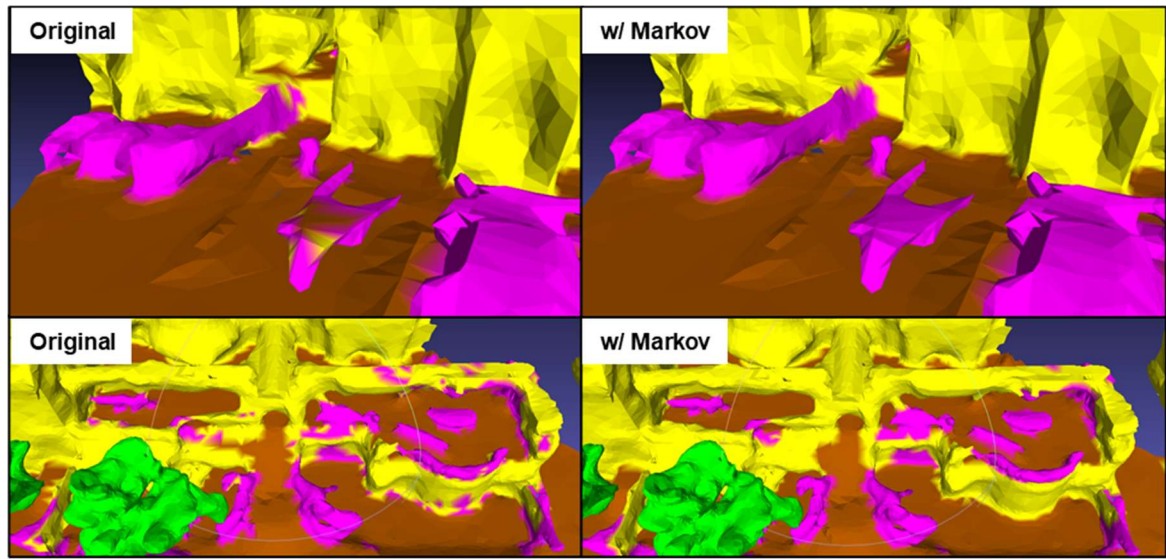

**Figure 19.** Visual impact of Markov context analysis on Xeos 3D tiles. The following parameters were used: $p = 1$, $\alpha_l = 1$, and $\beta = 0.5$.

The results presented in Table 3 reveal a significant impact of the contextual analysis on segmentation performance, with contrasting trends for different classes. More specifically, the contextual analysis shows a clear improvement in results for the "building" and "high_urban" classes when applied to vertices within semantic boundaries. However, an opposite trend emerges for vertices located at semantic boundaries, where the contribution of changes related to these vertices seems to deteriorate the overall segmentation results.

**Table 3.** IoU results for the different tests performed with the Markov context analysis algorithm according to different vertex groups.

| Algorithm Parameters ($\alpha_l = 1$; $p = 1$; $\beta = 0.5$) | IoU (%) | | | | | |
|---|---|---|---|---|---|---|
| | Mean IoU | Terrain | Vegetation | Building | High_urban | Unclassified |
| All vertices (328,600) | | | | | | |
| Original | 69.17 | 79.85 | 97.56 | 88.87 | 33.60 | 45.99 |
| With contextual analysis | 68.71 | 77.71 | 97.56 | 89.19 | 34.03 | 45.05 |
| Interior vertices (318,944) | | | | | | |
| Original | 71.61 | 85.96 | 98.22 | 91.80 | 35.14 | 46.92 |
| With contextual analysis | 71.66 | 85.36 | 98.24 | 92.13 | 35.94 | 46.62 |
| Boundary vertices (9656) | | | | | | |
| Original | 43.33 | 58.83 | 46.41 | 51.64 | 16.67 | 43.11 |
| With contextual analysis | 40.06 | 53.90 | 42.20 | 51.36 | 12.68 | 40.18 |

### 4.3. Results of Building Extraction with Cluster Analysis Algorithms

Tests were carried out on the KC15 tile from the simulated data, which contains 79,455 vertices classified as "building", as well as on the same tile as for the contextual analysis of the real data, which contains 40,877 vertices classified as "buildings". The former contains numerous isolated buildings of varying sizes, while the latter contains mainly attached buildings. Clustering analyses were performed using three algorithms: KMeans, spectral clustering, and DBSCAN. The results of these analyses are shown in Figures 20 and 21 for simulated and real data, respectively. The contextual analysis was integrated into the semantic segmentation process to address gaps, inconsistencies, and regional heterogeneity. This method fills the gaps and homogenizes the semantic regions. This improved segmentation guarantees data quality upstream of cluster analysis, ensuring

more accurate and meaningful results. It should be noted that the specific parameters used for each algorithm are not indicated, as the results illustrate a general example obtained by each of these algorithms.

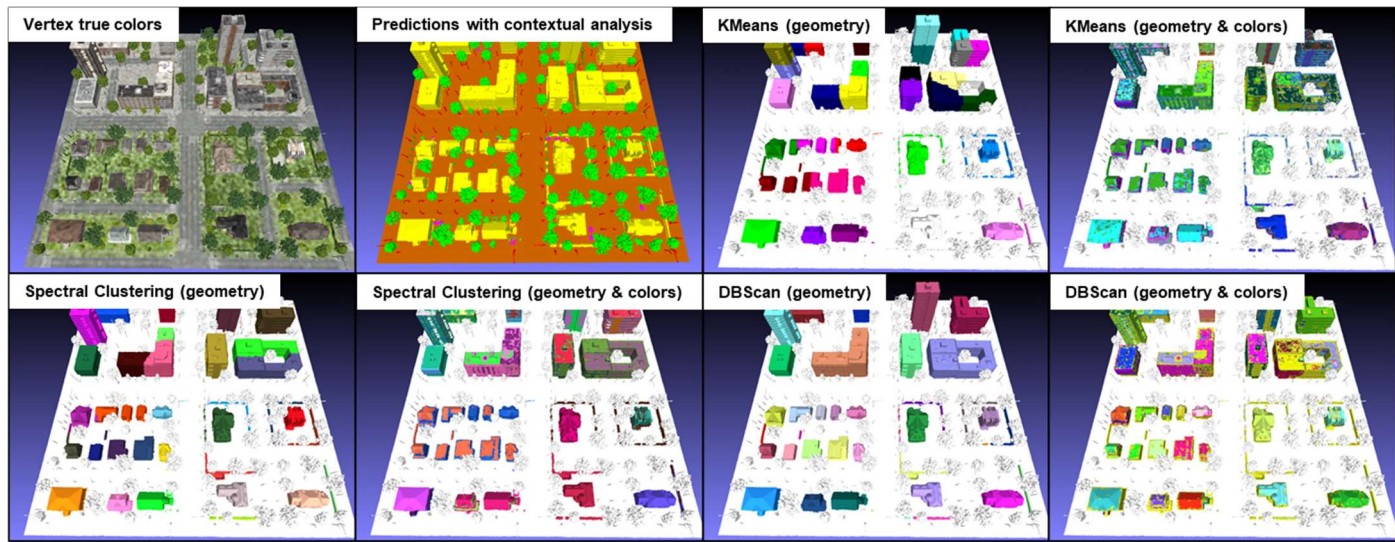

**Figure 20.** Results of the KMeans, spectral clustering, and DBSCAN algorithms on the simulated 3D tile KC15.

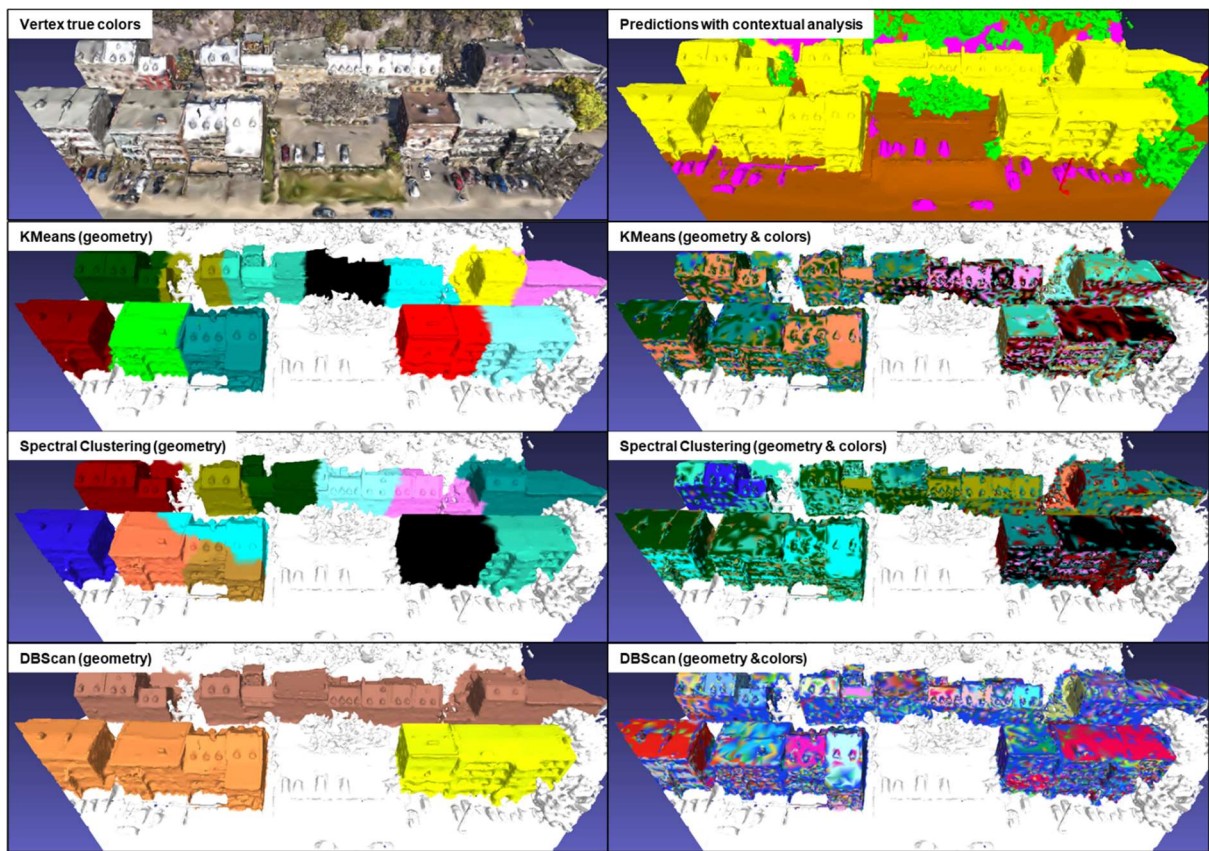

**Figure 21.** Results of KMeans, spectral clustering, and DBSCAN analyses on a validation tile from the Xeos 3D model.

For the KMeans algorithm using geometry alone, it should be noted that it tends to coarsely divide groups of vertices, leading to the merging of some isolated buildings into a single cluster. As for the KMeans algorithm using both geometry and color, when KMeans uses both geometric and color data, the clusters change considerably. This algorithm seems to be sensitive to variations in vertex color. However, the results more closely resemble a coarse semantic segmentation, seeking to roughly separate the different color variations present on buildings. A major drawback of this algorithm is that it requires a prior specification of the number of clusters to be created, making automation difficult. On the other hand, the other parameters are relatively simple to calibrate to obtain optimal results.

The results obtained with the spectral clustering algorithm are similar to those of KMeans, with a coarse division of buildings and sensitivity to variations in vertex color. It also suffers from the need to specify in advance the number of clusters to be created and involves other complex parameters to be calibrated to obtain optimal results.

The geometry-only DBSCAN algorithm produces interesting results. It efficiently extracts every single building or group of buildings, including hedges that had been misclassified as "building". However, DBSCAN that uses both geometry and color returns similar results to the other two methods, which also use both geometry and color and are not usable. A notable advantage of DBSCAN is that it does not require pre-specification of the number of clusters. However, it does require several initial trials to calibrate the "eps" and "min_samples" parameters in order to obtain optimal results for a specific dataset.

## 5. Discussion

The results of this study highlight several crucial aspects in the field of the semantic segmentation of 3D data, particularly for building segmentation. The use of mixed datasets, combining both real and simulated data, has proven to be a promising approach for improving segmentation performance. The results show that the model trained by transfer learning with simulated data outperformed the other models in terms of average IoU and IoU for the "building" class. However, it is important to note that the sensitivity of the model to the methods used to create the 3D mesh was highlighted, underlining the need for careful attention when preparing the data. Furthermore, the limited representation and errors generated during the Xeos 3D model creation process, such as the merging of certain poles and trees, are factors that can contribute to the model's difficulty in learning object features. This observation implies that texture may not consistently serve as a reliable parameter for learning object features in real data. NVIDIA recently developed Neuralangelo [60]. This innovative method combines the representational power of multi-resolution 3D hash grids with neural surface rendering. Neuralangelo brings significant improvements to the reconstruction of 3D structures from multi-view images by using digital gradients to compute higher-order derivatives and by progressively optimizing details on hash grids. Moreover, the utilization of 3D meshes characterized by improved geometry and texture holds the potential to mitigate ambiguity in object differentiation, as observed in this study, particularly when distinguishing between objects like poles and trees. This enhancement shows the prospect of delivering enhanced results in semantic segmentation. In addition, an improved version of PicassoNet, entitled PicassoNet++, was recently published by [61]. This new version performs slightly better than PicassoNet-II in the area of semantic segmentation.

Three-dimensional models generated synthetically by various artists and companies are typically designed with a high level of detail for applications such as animation, video game design, or architecture. When employing these models for data augmentation in a distinct context, it is frequently imperative to make substantial alterations to the 3D data, often involving the use of various software tools, as no single software or Python library comprehensively addresses all the associated challenges. This adjustment process can become quite intricate, especially when dealing with objects of diverse characteristics, further adding to the complexity of the task. Creating simulated data remains a laborious

and arduous task, raising the question of whether investing this valuable time might be better spent annotating a larger volume of real data.

The addition of a contextual analysis revealed advantages, particularly for improving building segmentation, although increasing the neighborhood level could lead to excessive overlap between semantic zones as well as a decrease in average IoUs. Additionally, the analysis excels at handling isolated vertices, which contributes to an overall improvement in results. However, an opposite trend emerges for vertices located at semantic boundaries, where the contribution of changes related to these vertices seems to deteriorate the overall segmentation results. Ultimately, it appears that the contextual analysis primarily serves a role in refining semantic class delineation rather than significantly enhancing quantitative outcomes.

With regard to cluster analysis, both the KMeans and spectral clustering algorithms often struggle to accurately segment isolated or non-isolated buildings. The DBSCAN algorithm using geometric data alone was found to be the most effective at extracting isolated buildings, but it presented difficulties with groups of connected buildings. Furthermore, this algorithm demands the identification of appropriate parameters to achieve precise individual building extraction. These parameter values may differ for each dataset, potentially introducing complexities in automating the process.

In summary, these results highlight the importance of carefully considering the combination of different approaches and techniques to achieve accurate and reliable segmentation in 3D meshes, particularly in scenarios where building complexity and data variability can represent significant challenges. Furthermore, it is worth noting the growing interest in instance segmentation techniques. These methods offer the potential to greatly enhance segmentation outcomes, particularly when they can leverage intrinsic texture information embedded within 3D mesh facets. This underscores the importance of staying abreast of developments in instance segmentation, exemplified by projects like Segment Anything [62], which push the boundaries of image segmentation models, and the Multi-View Stereo (MVS) building instance segmentation work [63], which demonstrates the possibilities of extracting 3D object instances in complex urban scenes. In addition, datasets like UrbanBiS [64], which offer rich annotations and encompass extensive urban areas, represent essential resources for benchmarking and advancing segmentation techniques. The combination of different approaches and their potential synergy, especially when harnessing texture image information in 3D mesh facets, presents exciting prospects for the future of 3D mesh segmentation.

## 6. Conclusions

In conclusion, the findings from this study shed light on the intricate and evolving field of semantic segmentation in 3D data, with a specific focus on building segmentation. The integration of mixed datasets, blending real and simulated data, exhibited promising results, especially when leveraging transfer learning. However, this study also underscored the critical importance of meticulous data preparation, given the model's sensitivity to 3D mesh quality. The discussion also emphasized the potential benefits of context analysis in refining class delineation and raised awareness about the challenges in cluster analysis, especially for isolated and non-isolated buildings. Ultimately, this study encourages a holistic approach that combines various techniques and stays abreast of instance segmentation developments, with a focus on leveraging texture information within 3D mesh facets. These collective efforts pave the way for the exciting future of 3D mesh segmentation, where innovation, benchmarking, and rich datasets are the key drivers of progress in this dynamic field.

**Author Contributions:** Conceptualization, Frédéric Leroux, Mickaël Germain, Étienne Clabaut and Yacine Bouroubi; methodology, Frédéric Leroux; validation, Frédéric Leroux, Mickaël Germain, Étienne Clabaut, Yacine Bouroubi and Tony St-Pierre; formal analysis, Frédéric Leroux; investigation, Frédéric Leroux; resources, Frédéric Leroux, Mickaël Germain, Étienne Clabaut, Yacine Bouroubi and Tony St-Pierre; data curation, Frédéric Leroux and Tony St-Pierre; writing—original draft preparation, Frédéric Leroux; writing—review and editing, Frédéric Leroux, Mickaël Germain, Étienne Clabaut, Yacine Bouroubi and Tony St-Pierre; visualization, Frédéric Leroux; supervision, Mickaël Germain, Étienne Clabaut, and Yacine Bouroubi; project administration, Mickaël Germain; funding acquisition, Mickaël Germain. All authors have read and agreed to the published version of the manuscript.

**Funding:** This research was undertaken, in part, thanks to funding from a Mitacs Accelerate grant (IT28137) and a Université de Sherbrooke grant (707582) held by Mickaël Germain.

**Data Availability Statement:** Data available on request due to restrictions privacy.

**Acknowledgments:** We would like to thank XEOS Imaging Inc. for letting us use some of their data. Additionally, we would like to express our gratitude to Van-Tho Nguyen for providing advisory counsel and technical expertise throughout this project.

**Conflicts of Interest:** The authors declare no conflicts of interest. The funders had no role in the design of the study; in the collection, analyses, or interpretation of data; in the writing of the manuscript; or in the decision to publish the results.

## Appendix A. Comparison of PicassoNet-II Results with Other Segmentation Models from Literature

| Method | OA | mAcc | mIoU | ceiling | floor | wall | beam | column | window | door | table | chair | sofa | bookcase | board | clutter |
|---|---|---|---|---|---|---|---|---|---|---|---|---|---|---|---|---|
| PointNet [59] | - | 49.0 | 41.1 | 88.8 | 97.3 | 69.8 | 0.1 | 3.9 | 46.3 | 10.8 | 58.9 | 52.6 | 5.9 | 40.3 | 26.4 | 33.2 |
| SEGCloud [95] | - | 57.4 | 48.9 | 90.1 | 96.1 | 69.9 | 0.0 | 18.4 | 38.4 | 23.1 | 70.4 | 75.9 | 40.9 | 58.4 | 13.0 | 41.6 |
| Tangent-Conv [57] | 82.5 | 62.2 | 52.8 | - | - | - | - | - | - | - | - | - | - | - | - | - |
| SPG [96] | 86.4 | 66.5 | 58.0 | 89.4 | 96.9 | 78.1 | 0.0 | **42.8** | 48.9 | 61.6 | 75.4 | 84.7 | 52.6 | 69.8 | 2.1 | 52.2 |
| PointCNN [68] | 85.9 | 63.9 | 57.3 | 92.3 | 98.2 | 79.4 | 0.0 | 17.6 | 22.8 | 62.1 | 74.4 | 80.6 | 31.7 | 66.7 | 62.1 | 56.7 |
| SSP+SPG [97] | 87.9 | 68.2 | 61.7 | - | - | - | - | - | - | - | - | - | - | - | - | - |
| GACNet [69] | 87.8 | - | 62.9 | 92.3 | 98.3 | 81.9 | 0.0 | 20.4 | 59.1 | 40.9 | 78.5 | 85.8 | 61.7 | 70.8 | 74.7 | 52.8 |
| SPH3D-GCN [18] | 87.7 | 65.9 | 59.5 | 93.3 | 97.1 | 81.1 | 0.0 | 33.2 | 45.8 | 43.8 | 79.7 | 86.9 | 33.2 | 71.5 | 54.1 | 53.7 |
| SegGCN [44] | 88.2 | 70.4 | 63.6 | 93.7 | **98.6** | 80.6 | 0.0 | 28.5 | 42.6 | **74.5** | 80.9 | 88.7 | 69.0 | 71.3 | 44.4 | 54.3 |
| MinkowskiNet [28] | - | 71.7 | 65.3 | - | - | - | - | - | - | - | - | - | - | - | - | - |
| KPConv [29] | - | 72.8 | 67.1 | 92.8 | 97.3 | 82.4 | 0.0 | 23.9 | 58.0 | 69.0 | 81.5 | **91.0** | **75.4** | **75.3** | 66.7 | 58.9 |
| DCM-Net [15] | - | 71.2 | 64.0 | 92.1 | 96.8 | 78.6 | 0.0 | 21.6 | **61.7** | 54.6 | 78.9 | 88.7 | 68.1 | 72.3 | 66.5 | 52.4 |
| PicassoNet-II (Prop.) | **90.4** | **75.7** | **69.8** | **94.4** | 98.1 | **85.1** | 0.0 | 33.8 | 59.5 | 80.9 | **82.8** | 90.0 | 79.3 | 74.9 | **70.3** | **58.1** |

**Figure A1.** PicassoNet-II performance on the S3DIS dataset. From [15].

| Method | mIoU | floor | wall | chair | sofa | table | door | cab | bed | desk | toil | sink | wind | pic | bkshf | curt | show | cntr | fridg | bath | other |
|---|---|---|---|---|---|---|---|---|---|---|---|---|---|---|---|---|---|---|---|---|---|
| SPLATNET$_{3D}$ [33] | 39.3 | 92.7 | 69.9 | 65.6 | 51.0 | 38.3 | 19.7 | 31.1 | 51.1 | 32.8 | 59.3 | 27.1 | 26.7 | 0.0 | 60.6 | 40.5 | 24.9 | 24.5 | 0.1 | 47.2 | 22.7 |
| Tangent-Conv [57] | 43.8 | 91.8 | 63.3 | 64.5 | 56.2 | 42.7 | 27.9 | 36.9 | 64.6 | 28.2 | 61.9 | 48.7 | 35.2 | 14.7 | 47.4 | 25.8 | 29.4 | 35.3 | 28.3 | 43.7 | 29.8 |
| PointCNN [68] | 45.8 | 94.4 | 70.9 | 71.5 | 54.5 | 45.6 | 31.9 | 32.1 | 61.1 | 32.8 | 75.5 | 48.4 | 47.5 | 16.4 | 35.6 | 37.6 | 22.9 | 29.9 | 21.6 | 57.7 | 28.5 |
| PointConv [31] | 55.6 | 94.4 | 76.2 | 73.9 | 63.9 | 50.5 | 44.5 | 47.2 | 64.0 | 41.8 | 82.7 | 54.0 | 51.5 | 18.5 | 57.4 | 43.3 | 57.5 | 43.0 | 46.4 | 63.6 | 37.2 |
| SPH3D-GCN [18] | 61.0 | 93.5 | 77.3 | 79.2 | 70.5 | 54.9 | 50.7 | 53.2 | 77.2 | 57.0 | 85.9 | 60.2 | 53.4 | 4.6 | 48.9 | 64.3 | 70.2 | 40.4 | 51.0 | 85.8 | 41.4 |
| KPConv [29] | 68.4 | 93.5 | 81.9 | 81.4 | 78.5 | 61.4 | 59.4 | 64.7 | 75.8 | 60.5 | 88.2 | 69.0 | 63.2 | 18.1 | 78.4 | 77.2 | 80.5 | 47.3 | 58.7 | 84.7 | 45.0 |
| SegGCN [44] | 58.9 | 93.6 | 77.1 | 78.9 | 70.0 | 56.3 | 48.4 | 51.4 | 73.1 | 57.3 | 87.4 | 59.4 | 49.3 | 6.1 | 53.9 | 46.7 | 50.7 | 44.8 | 50.1 | 83.3 | 39.6 |
| MinkowskiNet [28] | **73.6** | 95.1 | **85.2** | **84.0** | 77.2 | **68.3** | **64.3** | 70.9 | **81.8** | **66.0** | 87.4 | 67.5 | **72.7** | 28.6 | **83.2** | **85.3** | **89.3** | **52.1** | **73.1** | **85.9** | **54.4** |
| DCM-Net [15] | 65.8 | 94.1 | 80.3 | 81.3 | 72.7 | 56.8 | 52.4 | 61.9 | 70.2 | 49.4 | 82.6 | 67.5 | 63.7 | **29.8** | 80.6 | 69.3 | 82.1 | 46.8 | 51.0 | 77.8 | 44.9 |
| PicassoNet-II (Prop.) | 69.6 | **95.6** | 84.8 | 83.7 | **79.9** | 61.9 | 61.5 | **70.9** | 79.0 | 54.3 | **90.8** | **70.3** | 70.0 | 25.0 | 78.7 | 81.5 | 79.0 | 45.9 | 55.1 | 70.4 | 52.9 |

**Figure A2.** PicassoNet-II performance on the ScanNet dataset. From [15].

## Appendix B. Compilation of Selected Textured 3D Tiles from the Xeos 3D Model

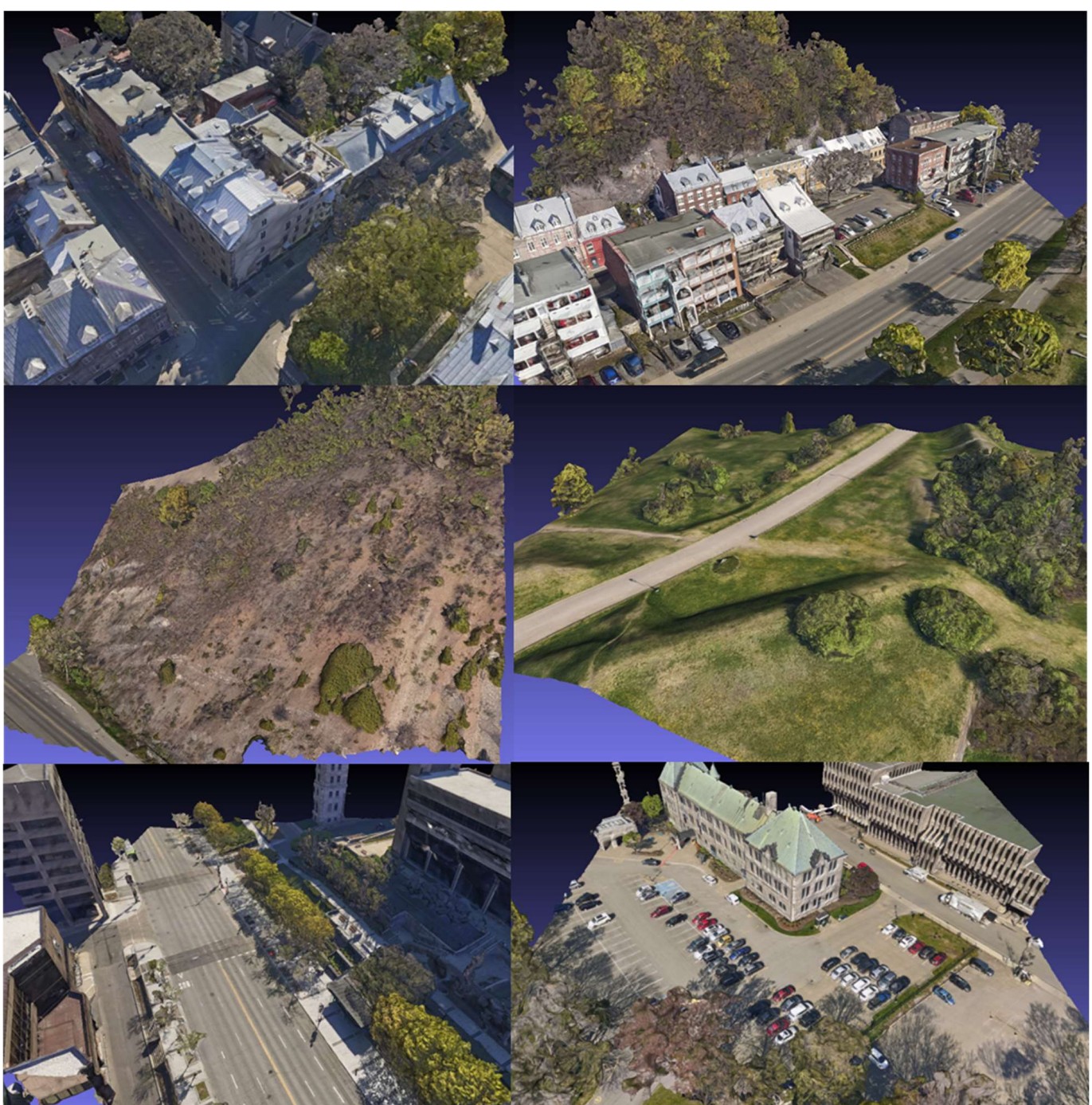

**Figure A3.** Compilation of selected textured 3D tiles from the Xeos 3D model.

## Appendix C. Compilation of Annotated Tiles from the Xeos 3D Model

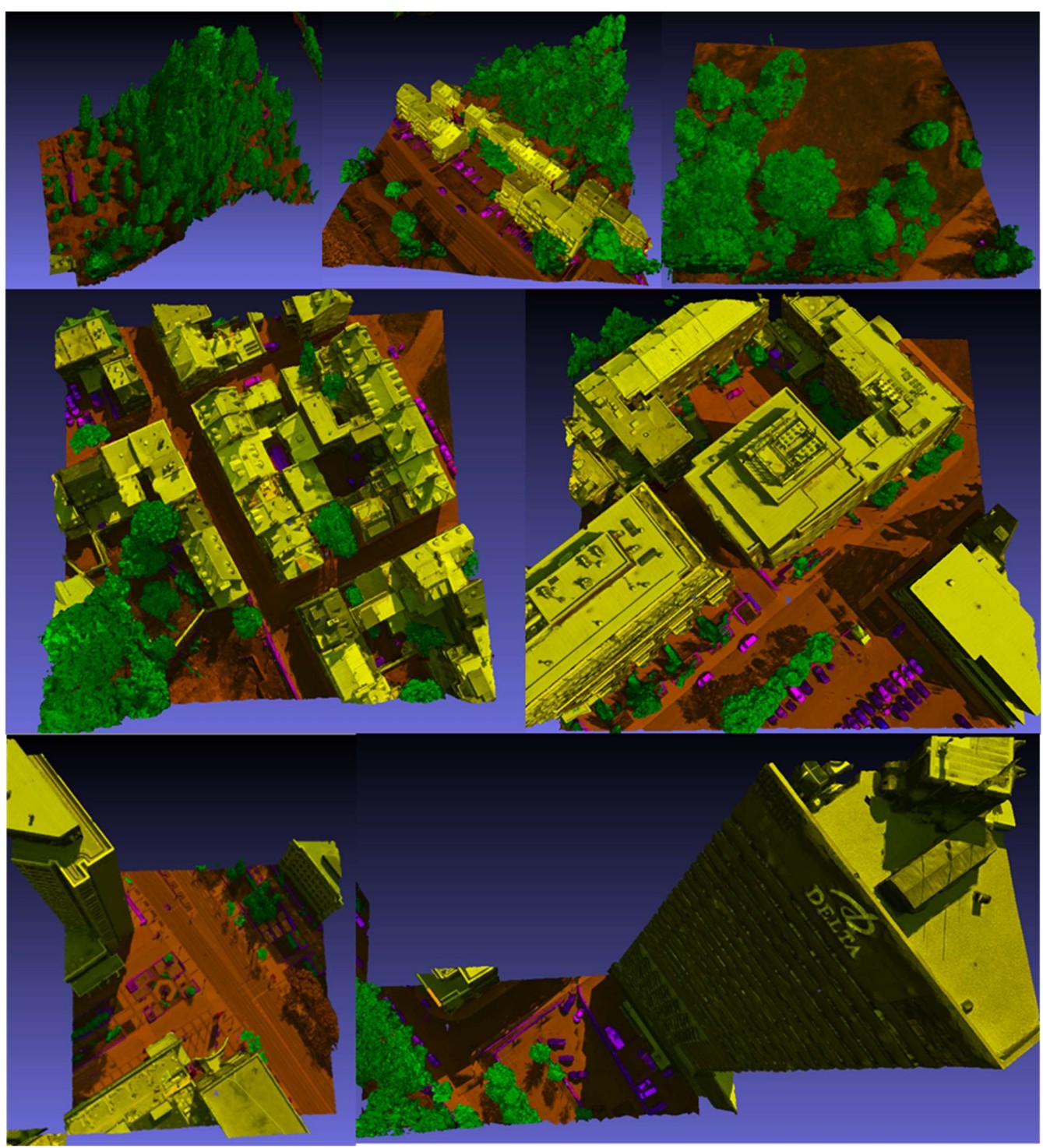

**Figure A4.** Compilation of annotated tiles from the Xeos 3D model.

## Appendix D. Inventory of 3D Objects from Epic Games Market and Evermotion

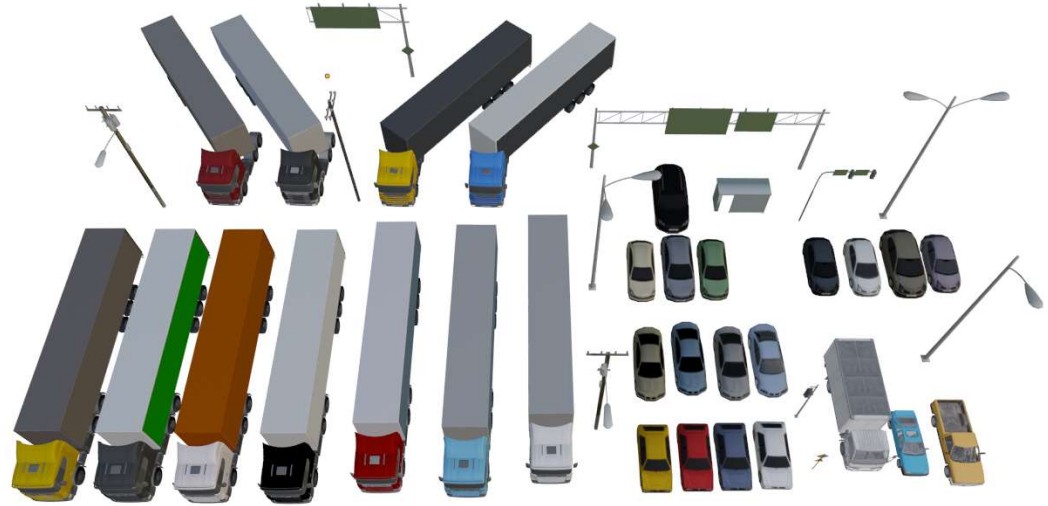

**Figure A5.** Inventory of 3D objects from Epic Games Market and Evermotion.

## Appendix E. Overview and Distribution of Simulated Tiles in the Training Dataset

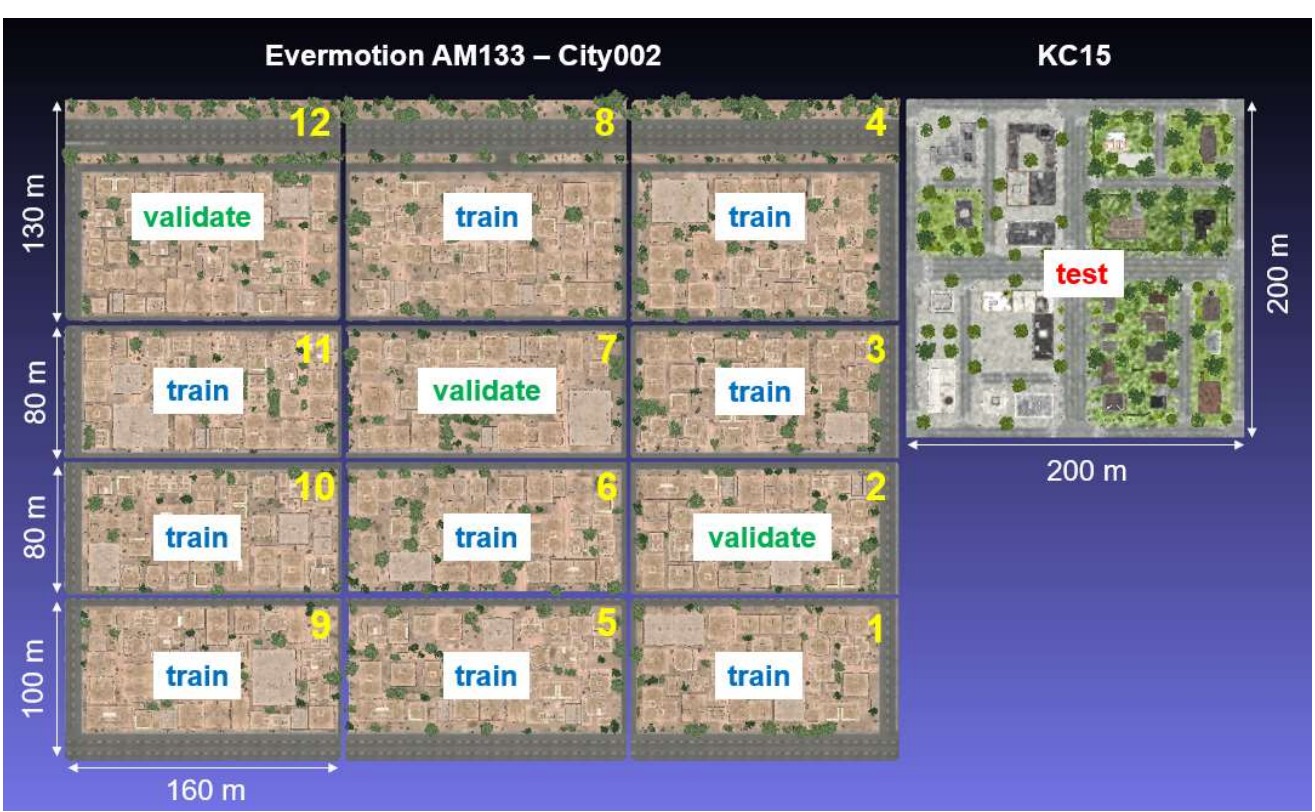

**Figure A6.** Overview and distribution of simulated tiles from Evermotion model AM133-City002 and KC15.

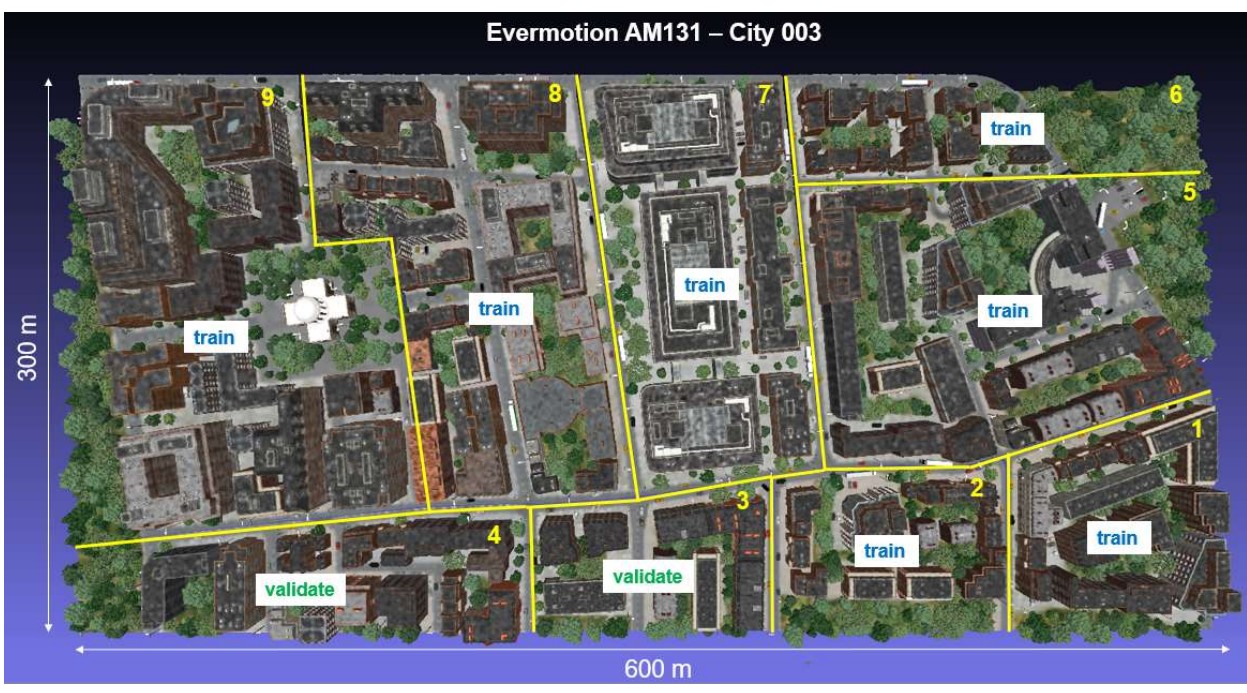

**Figure A7.** Overview and distribution of simulated tiles from Evermotion model AM131-City003.

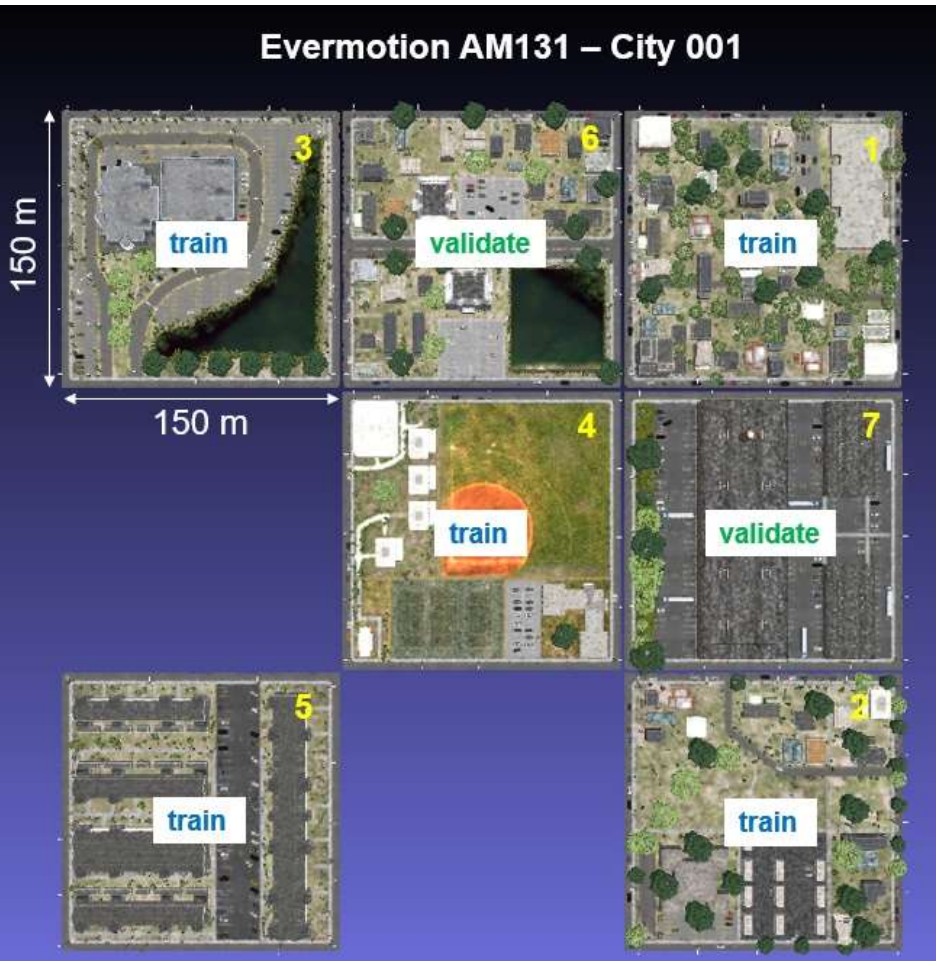

**Figure A8.** Overview and distribution of simulated tiles from Evermotion model AM131-City001.

## Appendix F. Overview of the SUM-Helsinki Dataset

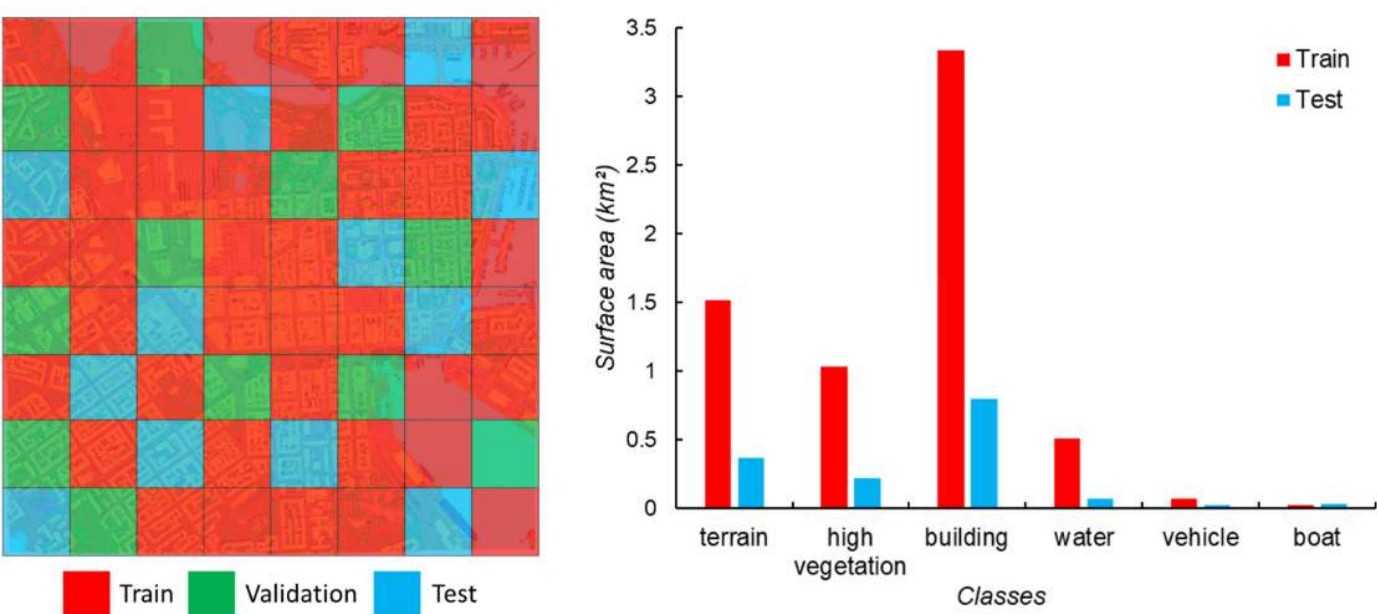

**Figure A9.** Distribution of SUM-Helsinki tiles during training (**left**). Area covered by each class in terms of surface area (**right**). From [28].

## Appendix G. Charts Illustrating the Progression of Precision and Loss of Drive in Each Era for the Different Drives

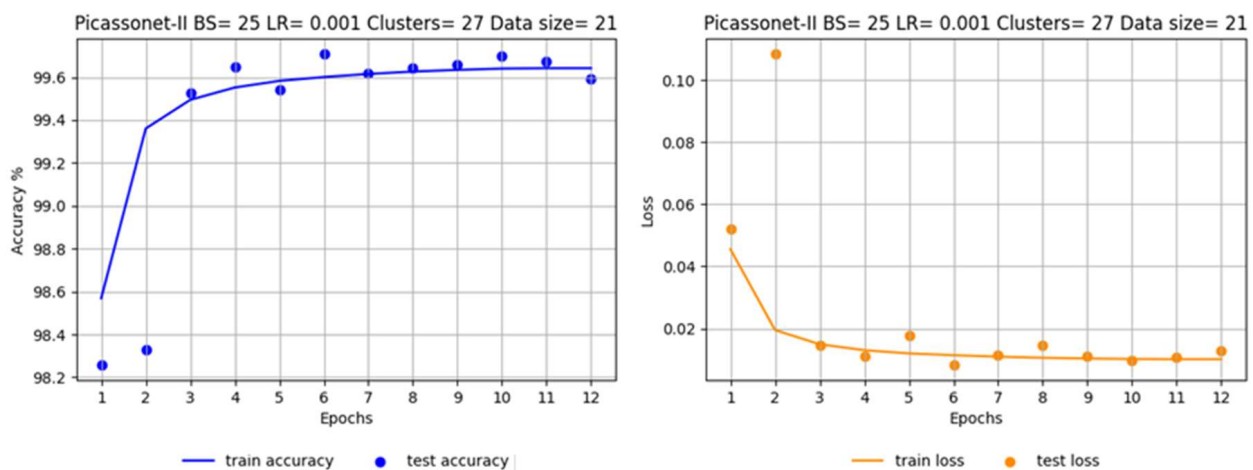

**Figure A10.** Progression of accuracy (**left**) and loss at each epoch (**right**) for training on simulated data only.

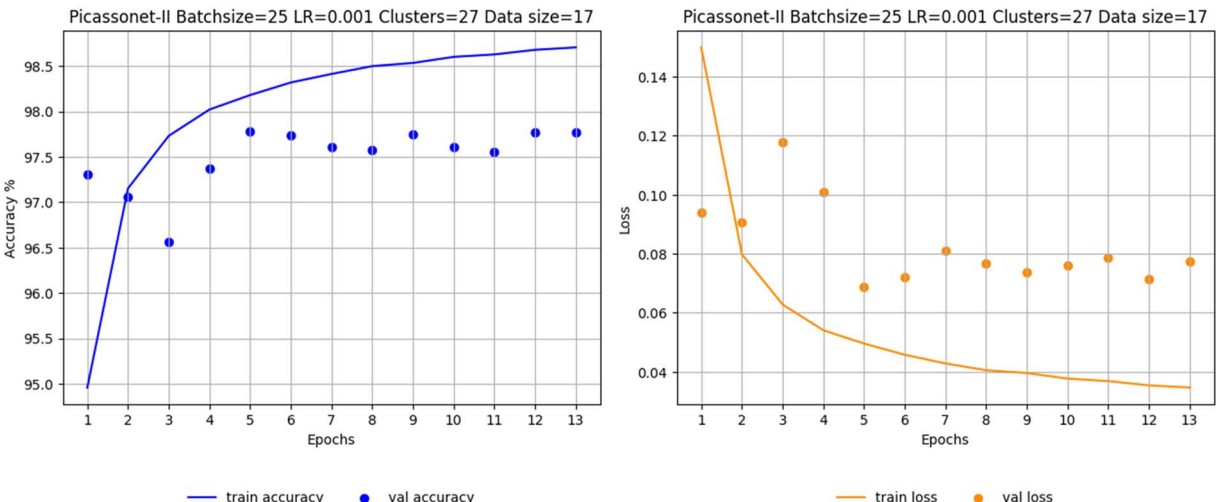

**Figure A11.** Progression of accuracy (**left**) and loss at each epoch (**right**) for training on real data only.

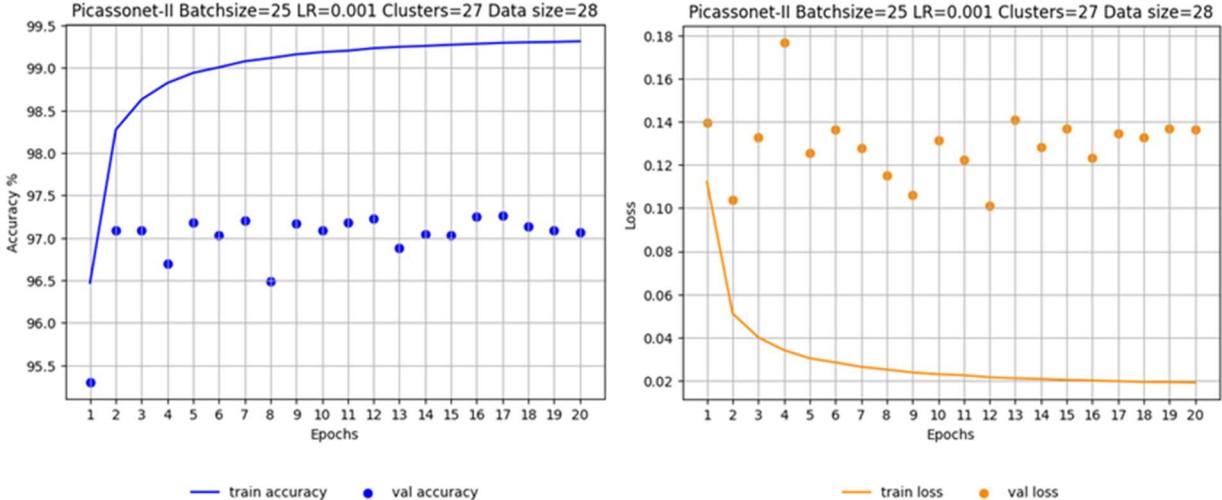

**Figure A12.** Progression of accuracy (**left**) and loss at each epoch (**right**) for training on real and simulated data.

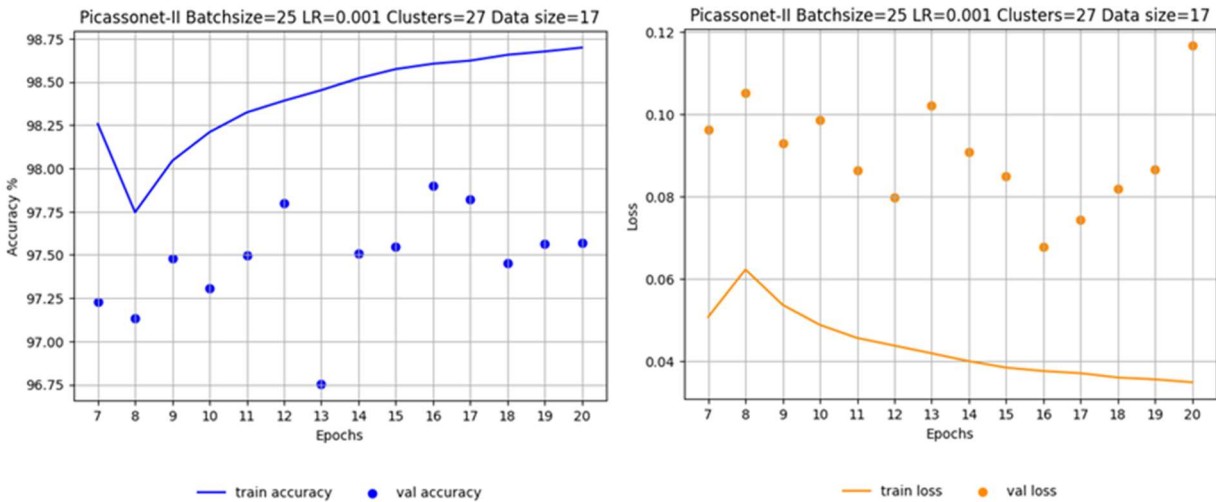

**Figure A13.** Progression of accuracy (**left**) and loss at each epoch (**right**) for training on real data by transfer learning.

## Appendix H. Compilation of Inferences Generated with PicassoNet-II (Transfer Learning Model)

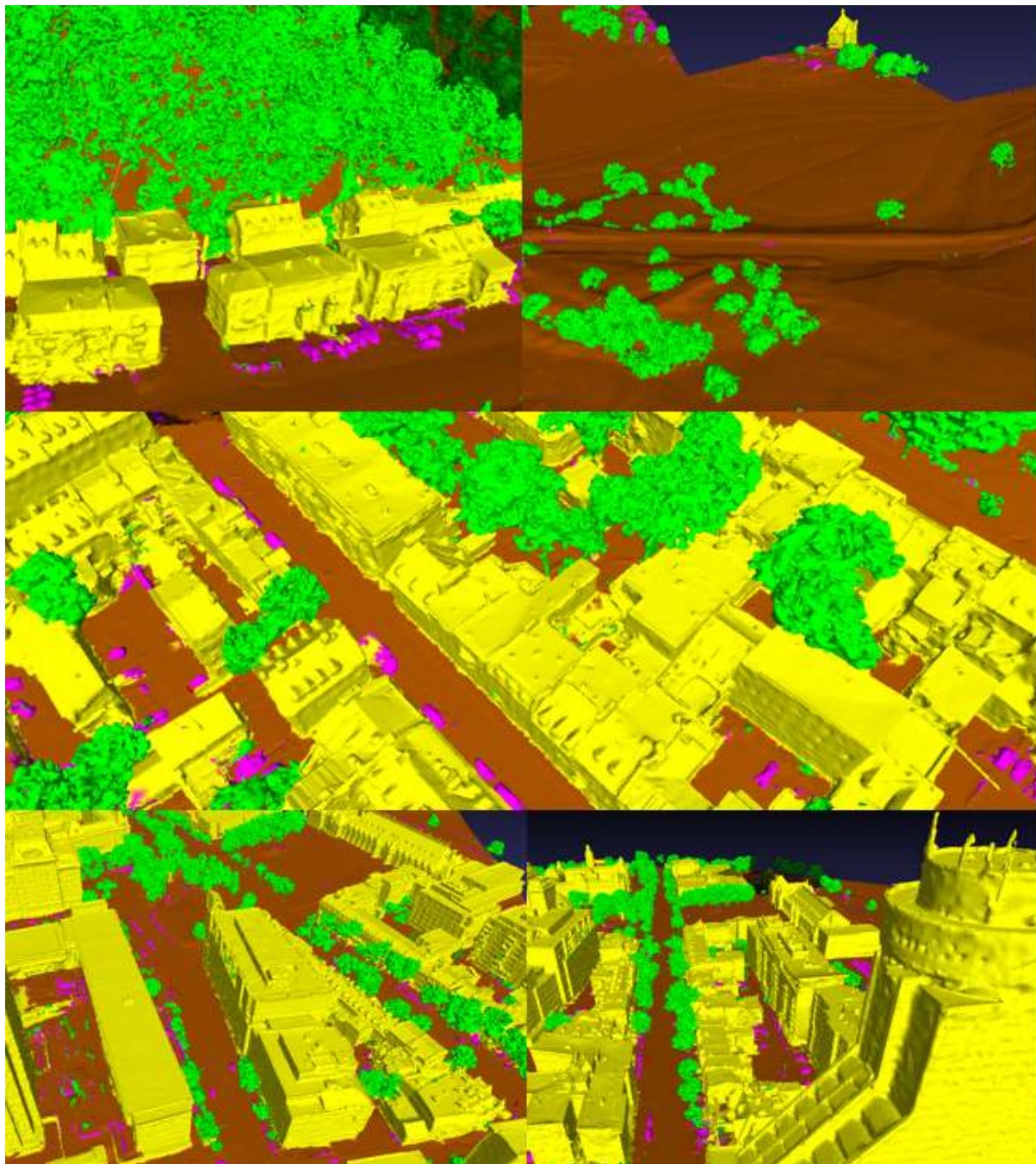

**Figure A14.** Compilation of inferences generated with PicassoNet-II (transfer learning model). Shaded areas correspond to annotated data.

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
