# Peer review of "Improving Three-Dimensional Building Segmentation on Three-Dimensional City Models through Simulated Data and Contextual Analysis for Building Extraction"

_ijgi, doi:10.3390/ijgi13010020_

Round 1

Reviewer 1 Report

Comments and Suggestions for Authors

Format

1、   The Materials and Methods section and its subsections are incorrectly numbered.

2、   Similar to the table at line 573, many tables have problems with line numbers overlapping with table content.

3、   The figure at line 793 is too blurry.

Content

1、   The paragraph at line 138 does not particularly highlight the advantages of real and simulated datasets, limiting itself too much to a textual description of the application of both.

2、   At line 214, what are the "5 usual files"? Please explain.

3、   The paragraph at line 383 should logically be placed before line 382.

4、   Why does the paragraph at line 481 suddenly talk about the inference results on Xeos tiles? This subsection is supposed to be about training and inference on simulated data only.

5、   At line 519, are the 3 tiles used for validation real or simulated data? Please explain.

6、   Why was the experiment at line 582 performed only on the Xeos dataset and not on both real and simulated data?

7、   The reference to Figure A1 should be [15], not [12].

8、   Why does Figure H1 only show the inference results without attaching corresponding ground truth?

Reviewer 2 Report

Comments and Suggestions for Authors

The paper called "IMPROVING 3D BUILDING SEGMENTATION ON 3D CITY 2 MODELS THROUGH SIMULATED DATA AND CONTEX-3 TUAL ANALYSIS FOR BUILDING EXTRACTION" is very well written. It clearly states the contemporary state-of-the-art of the 3D segmentation field of research and appropriately uses different methods to prove that their method has advantages over different ones. Nevertheless, the authors critically evaluate their approach and add limitations to their solution. The text is well structured, and references are used appropriately. English is brilliant and easy to understand, even for non-native speakers. The research value of this contribution is very high as well.

I have just a few remarks here:

Line 62. The comma is missing at the end of the line, resp. sentence.
Line 149. The comma is missing after citations at the beginning of the sentence.
Figures 14, 15, and 16: The caption text of the figures is in French, not English.

And I have two questions for the authors as well:

Page 5: in the text before Figure 2, there is information about the size of the tiles, particularly 55x55 meters. In Figure 2, there are tiles shown, and their size is depicted, but it is written that a particular tile is 56x56 meters in size. Is it a different size of a tile (while in Figure 2, there are many non-standard sizes of the tiles), or is it the one that should have a size of 55x55 meters?

Line 208: the annotation process takes between 1h30 and 4h30 for the given dataset. What are the specifications of the hardware used for such a process?

Reviewer 3 Report

Comments and Suggestions for Authors

The title clearly describes what the author wants to convey.

The abstract needs to be improved so that readers can more quickly understand the results of your research.

The introduction already has sufficient references.

The data and analysis are very complete. These results help strengthen future research, especially in the construction of buildings in the city.

Reviewer 4 Report

Comments and Suggestions for Authors

The paper topic is very interesting and presents many points of originality and innovation.   The paper presents a robust analysis, supported by data, and results compatible with the objectives.

The paper is well written, and the study is well thought, supported by a substantial scientific literature. The model is interesting and lends itself to further study in the future. For this reviewer, the paper can be published as is, being sufficiently complete in all its parts.

Author Response

Comments:

The paper topic is very interesting and presents many points of originality and innovation.   The paper presents a robust analysis, supported by data, and results compatible with the objectives.

The paper is well written, and the study is well thought, supported by a substantial scientific literature. The model is interesting and lends itself to further study in the future.

Response to Comments:

Thank you for your positive feedback on our paper! We appreciate your constructive comments!

Round 2

Reviewer 1 Report

Comments and Suggestions for Authors

The author's responses to the initial review comments are relevant and well developed, as well as persuasive. However, I think the author still needs to improve on the following two points.

The first is Section 2.4, where I think the author still fails to make a strong enough argument by citing existing work to illustrate the section theme as he did in Section 2.1, Section 2.2, and Section 2.3.

Next is the second comment in the initial review comments regarding content. Line 222 of the latest version of the manuscript also needs to be revised, i.e., it also needs to be explicitly stated that details about these 5 usual files will be described in the 'Preparation of the real dataset' section.

In summary, I believe that there is still some room for improvement in the latest version of the manuscript, and I look forward to further improvements from the author.
